# A zinc oxide resonant nano-accelerometer with ultra-high sensitivity

Pengfei Xu [1], Dazhi Wang [1,2,3,4] ✉, Jianqiao He[1], Yichang Cui[1], Liangkun Lu[1], Yikang Li[1], Xiangji Chen[1], Chang Liu[1], Liujia Suo[1], Tongqun Ren[1], Tiesheng Wang [1] & Yan Cui[1,2]

Nanoelectromechanical system accelerometers have the potential to be utilized in next-generation consumer electronics, inertial navigation, and seismology due to their low cost, small size, and low power consumption. There is an urgent need to develop resonant accelerometer with high sensitivity, precision and robustness. Here, a zinc oxide resonant nano-accelerometer with high sensitivity has been designed and prototyped using zinc oxide nanowires. Within a device two nanowires were symmetrically placed close to a notched flexure to evaluate acceleration based on differential resonant frequencies. Additionally, microleverages were integrated in the accelerometer to enhance its sensitivity by amplifying the inertial force. High performance of the accelerometer has been demonstrated by the measured absolute sensitivity (16.818 kHz/g), bias instability (13.13 µg at 1.2 s integration time) and bandwidth (from 4.78 to 29.64 kHz), respectively. These results suggest that zinc oxide nanowires could be a candidate to develop future nanoelectromechanical resonant accelerometer potentially used for inertial navigation, tilt measurement, and geophysical measurements.

Microelectromechanical systems (MEMS) accelerometers have been extensively used in the geophysics[1–3], consumer electronics[4–7], and navigation systems[8,9] due to their small size, low power consumption, and high reliability. It is, however, still challenging to build high performance accelerometer to achieve high sensitivity and stability[10,11]. Both noise reduction and sensitivity enhancement to external acceleration are required for such high performance device. Nanoelectromechanical systems (NEMS) resonant accelerometer converts the external acceleration to a modulated frequency output by changing the stiffness of a resonator which is compatible to integrated circuits and high resistant to external electronic noises[12–16]. NEMS resonant accelerometer is expected to have a strong competence in high precision acceleration measurement owing to its ultrasmall mass, high resonant frequency and ultralow power consumption[17,18].

In recent years, various materials and structures have been employed to enhance the sensitivity and stability of M/NEMS resonant

accelerometers. A concept of micromechanical silicon oscillating accelerometer (SOA) was proposed and an in-plane vibration electrostatically driven resonant accelerometer was successfully developed with a resonant frequency of 20 kHz and a scale factor of 100 Hz/g using silicon-on-glass (SOG) processes by refs. 9,19. Comi et al. have developed a two-axis single-stage lever resonant accelerometer with a resonant frequency of 15 kHz and a sensitivity of 250 Hz/g[20,21]. To obtain higher stability and sensitivity, Pandit et al. have designed and demonstrated a single-stage amplified differential resonant accelerometer with a resonant frequency exceeding 100 kHz and a differential acceleration scale factor of 5.61 kHz/g for high precision acceleration[22–24]. Benefitting from nanomanufacturing technologies, the application of nano-resonators has promoted the resolution of the accelerometer to overcome the limitation of resolution-bandwidth-footprint trade-off in traditional technologies. Such accelerometer has been developed with a scale factor up to 1.4 MHz/g and a broad

[1]Key Laboratory for Micro/Nano Technology and System of Liaoning Province, Dalian University of Technology, Dalian 116024, China. [2]State Key Laboratory of High-performance Precision Manufacturing, Dalian University of Technology, Dalian 116024, China. [3]Ningbo Institute of Dalian University of Technology, Ningbo 315000, China. [4]Liaoning Huanghai Laboratory, Dalian University of Technology, Dalian 116024, China. ✉e-mail: d.wang@dlut.edu.cn

bandwidth of 1.5 kHz by refs. 25,26. The doped Si nano-resonator is, however, vulnerable to oxidation which will be inevitably detrimental for long-term application even with vacuum packaging. Villanueva et al. have raised a resonant NEMS accelerometer based on graphene nano-resonator which happens to resolve this issue with a sensitivity of approximately 4.41 kHz/g and a resonant frequency of 40 kHz[27]. However, the resonant frequency measurements of the graphene devices were performed by using a laser Doppler vibrometer (LDV) pointing at the device and excited by a piezo-shaker, which were not compatible for integration with electronic circuits.

Here, a zinc oxide (ZnO) resonant nano-accelerometer using a crystalline ZnO nanowire as a nano-resonator has been developed with high sensitivity. Compared to other resonant accelerometers, the resonant accelerometer based on ZnO nanowires can be excited and detected simultaneously, thereby facilitating integration with electronic circuits. Moreover, the crystalline ZnO nanowires with wurtzite structure were totally oxidized, which facilitates long term application due to the high resistant to oxidation[28,29]. As a foundation for its high performance, a pair of ZnO nanowires were fixed on the proof mass which swung around a notched flexure when subjected to external acceleration. This leads to a change in axial stress in the symmetrically distributed ZnO nanowires resulting in a shift in the resonant frequency. Compared to the single resonator design, the prototyped device evaluated the acceleration with a differential architecture, which could eliminate the majority of the resonant frequency shifts caused by temperature changes intrinsically. Furthermore, the microleverage was employed to improve the sensitivity of the accelerometer by amplifying the inertial force. The ZnO resonant nano-accelerometer was prepared based on the silicon-on-insulator (SOI) machining process and the ZnO nano-resonator with a diameter of nanoscale and a length of microscale was prepared by optical microscopy nanomanipulation technique and the focused ion beam (FIB) technique. The sensitivity of the accelerometer was measured to be 16.818 kHz/g with bandwidth from 4.78 to 29.64 kHz by the mixing principle and a series of tilt tests by rotating the device in the range of ±90 degrees (accelerations of ±1 g). Meanwhile, the Allan deviation yields a bias instability of 13.13 µg at 1.2 s integration time.

## Results
### Structure of accelerometers
The ZnO nano-resonant accelerometer consists of four parts: a proof mass as the motivating element, two ZnO nano-resonators as the detection element, a microleverage mechanism to amplify the axial force and a support beam with a notched flexure, as illustrated in Fig. 1. A single-point anchor with unloading grooves was adopted to reduce external stress coupled to the ZnO nano-resonator (detailed information was provided in Supplementary Materials Note S4). The support beam of the accelerometer was designed to be stiff in all directions except bending in the sensing direction, for which a notched flexure was incorporated. If the notched flexure was too long, the stability and resistance to overload of the accelerometer would be compromised. The width of the notched flexure, as a key parameter of the accelerometer motivating component, was crucial to improving the sensitivity of the accelerometer. To achieve higher sensitivity and stability, the size of the notched flexure was designed and optimized through finite element analysis (FEA, detailed information was provided in Supplementary Materials Note S2). Considering the performance of the accelerometer and the fabrication process, a rectangle notched flexure with a width of 3 µm and a length of 2 µm was proposed. To enhance the sensitivity of the accelerometer, a microleverage mechanism was implemented. According to the position of the pivot beam, input beam, and output beam of the microleverage, the microleverage was classified into three types. Herein, the first type of microleverages was adopted to improve the sensitivity of the accelerometer and facilitate electrodes fabrication. The structure of the

microleverage was optimized with FEA method which evaluated its performance by the deformation of the ZnO nano-resonators (detailed information was provided in Supplementary Materials Note S3). Herein, the support beam of the microleverage was determined to be 3 µm with a distance of 36 µm from the output beam.

After conducting the optimization described above, the parameters of ZnO resonant nano-accelerometer were listed in Table 1. The proof mass of the ZnO resonant nano-accelerometer swings around the notched flexure under the external acceleration. Thus, the stiffness of ZnO nano-resonators symmetrically distributed about the notched flexure were changed by the axial force. This acquired the proof mass to separate from the handle layer and buried silicon dioxide (BOX) to form a movable structure. Hence, the fabrication process based on SOI wafer and crystalline ZnO nanowires was proposed for ZnO resonant nano-accelerometers, which was displayed in Fig. 1c. A 4 in SOI wafer with 25 µm thick device layer was prepared as the starting substrate. First, a trench with a depth of 4 µm was etched with reactive ion etching (RIE) for gate electrodes. Then, 1 µm of silicon oxide was grown with chemical vapor deposition (CVD) as electrical isolation. Metal electrodes of 20 nm chromium (Cr) and 0.3 µm gold (Au) were grown and patterned on the structure layer with lift-off technique to actuate and detect the resonant behavior of the ZnO nano-resonators. After that, the structure layer of SOI wafer was patterned with deep reactive ion etching (DRIE) for the proof mass of the ZnO resonant nano-accelerometer. Finally, the handle layer and BOX were removed by DRIE to release the proof mass, as displayed in Fig. 1g. The ZnO nano-resonator was formed based on crystalline ZnO nanowires, which was prepared using optical microscope nanomanipulation technique and FIB technique. A pair of ZnO nanowires were symmetrically distributed about the notched flexure by optical microscope nanomanipulation, as shown in Fig. 1d. Then, Pt atoms were deposited on the interface between the ZnO nanowires and the electrodes to fix the ZnO nanowires forming a double clamped beam structure and improve contact behavior between ZnO nanowires and electrodes, which were displayed in Fig. 1e, f. To actuate the ZnO nano-resonator into high frequency vibration and detect its resonant behavior electrically, the accelerometer was wire bonded and installed in a ceramic package.

### Sensitivity of the ZnO resonant nano-accelerometer
For ZnO resonant nano-accelerometers, the axial force sensitivity of ZnO nano-resonators was crucial for the high performance resonant accelerometers. The ZnO nano-resonators were symmetrically distributed on both sides of the notched flexure, and it was desirable for the two ZnO nano-resonators to have the same shape and size to maximize the effect of differential architecture detection. The resonant frequency of the ZnO nano-resonator under different acceleration, $f_{res}$, was described by Eq. (1), where $f_O$ was the resonant frequency of the ZnO nano-resonator without acceleration, $m$ was the proof mass, $a$ was the acceleration, $I$ was the moment of inertia, $d$ was the diameter, $L$ was the length, $\rho$ was the density and $E$ was Young modulus of the ZnO nanowire, respectively. The resonant frequency differential of two ZnO nano-resonators to the acceleration can be expanded as Eq. (2), which indicates the resonant frequency shifts of the resonant accelerometer is proportional to the acceleration applied on the proof mass[30]. Furthermore, the diameter and the length of the nanowire play great effect on the axial force sensitivity (detailed information was provided in Supplementary Materials Note S1). To analyze the relationship between the accelerometer performance and the ZnO nanowire size, a single nano-resonator was studied using FEA. The first order resonant mode was used to evaluate the resonant frequency shift caused by the external acceleration. The relationship between resonant frequency, resonant frequency shift caused by the axial force and the nanowire size was illustrated in Fig. 2b. The resonant frequency of the ZnO nano-resonator increases linearly with the diameter of the

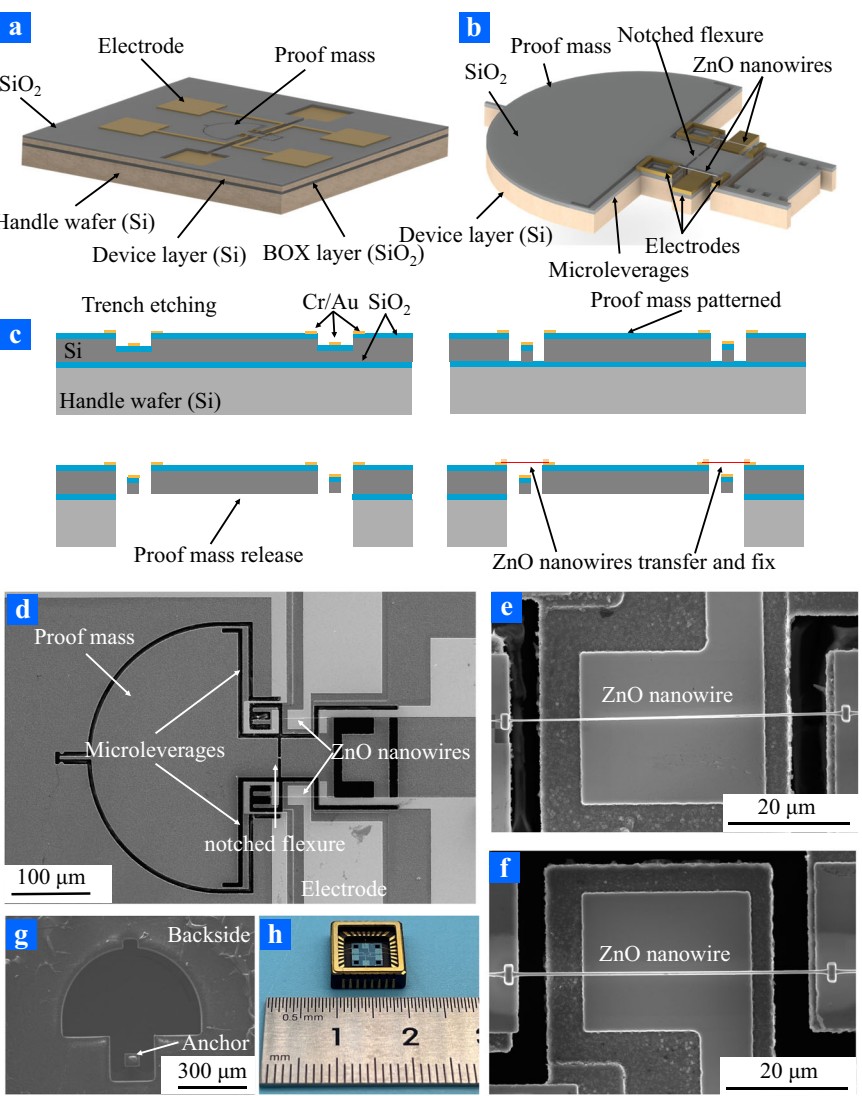

**Fig. 1 | Design and fabrication of ZnO resonant nano-accelerometers. a** Three-dimensional overview. **b** Detailed schematics showing. **c** Fabrication flow processes. The trenches were etched in the 25 μm silicon device layer of SOI substrate and depositing Cr/Au electrodes on the surface after SiO$_2$ deposited by chemical vapor deposition (CVD). The accelerometer structure was etched with deep reactive ion etching (DRIE) to form the silicon proof mass and microleverages. The silicon proof mass was released by etching handle layer and buried silicon dioxide (BOX) layer with DRIE methods. The ZnO nanowires were transferred with optical microscope nanomanipulation technique and fixed with FIB technique. **d** The scanning electron microscope (SEM) image of the accelerometer structure. **e, f** SEM images of the ZnO nanobeam. **g** The SEM image of the backside of the accelerometer. **h** The wire-bonded and installed ZnO resonant nano-accelerometer.

resonant nanowire (DNR) and decreases with the length of the resonant nanowire (LNR). Furthermore, the resonant frequency shifts caused by axial force decreases rapidly with the increase of diameter and increases with the increase of length. Notably, when the diameter of the nanowire is smaller than 500 nm, the resonant frequency shifts of the ZnO nano-resonator decrease significantly. Therefore, reducing the diameter of the nanowire is a convincing approach to improve the sensitivity of the accelerometer. Hence, the ZnO nano-resonant accelerometer with a pair of symmetrically distributed ZnO nano-resonators based on crystalline ZnO nanowires has the potential to surpass the resonator size to exceed microscale and greatly improve the sensitivity of the resonant accelerometer.

$$\begin{cases} f_{res} = \dfrac{4.73^2}{2\pi}\sqrt{1 \pm \dfrac{0.749 maL^2}{Ed^4}} \\ f_0 = \dfrac{4.73^2 d}{2\pi L}\sqrt{\dfrac{5E}{96\rho}} \end{cases} \tag{1}$$

$$\Delta f = 1.91 \frac{maL}{Ed^3}\sqrt{\frac{E}{\rho}} \tag{2}$$

The sensitivity of the accelerometer was analyzed with FEA method, and the results were displayed in Fig. 2. The working mode of

**Table 1 | Structural parameters of the ZnO resonant nano-accelerometer**

| Dimension | Value |
|---|---|
| Length of ZnO nanobeam | 50 μm |
| Diameter of ZnO nanobeam | 550 nm |
| Proof mass | 9.287 μg |
| Pivot beam (length × width) | 30 × 3 μm |
| Width of microleverage | 8 μm |
| Width of output beam | 5 μm |
| Support beam (length × width) | 100 × 55 μm |
| Notched flexure (length × width) | 3 × 2 μm |

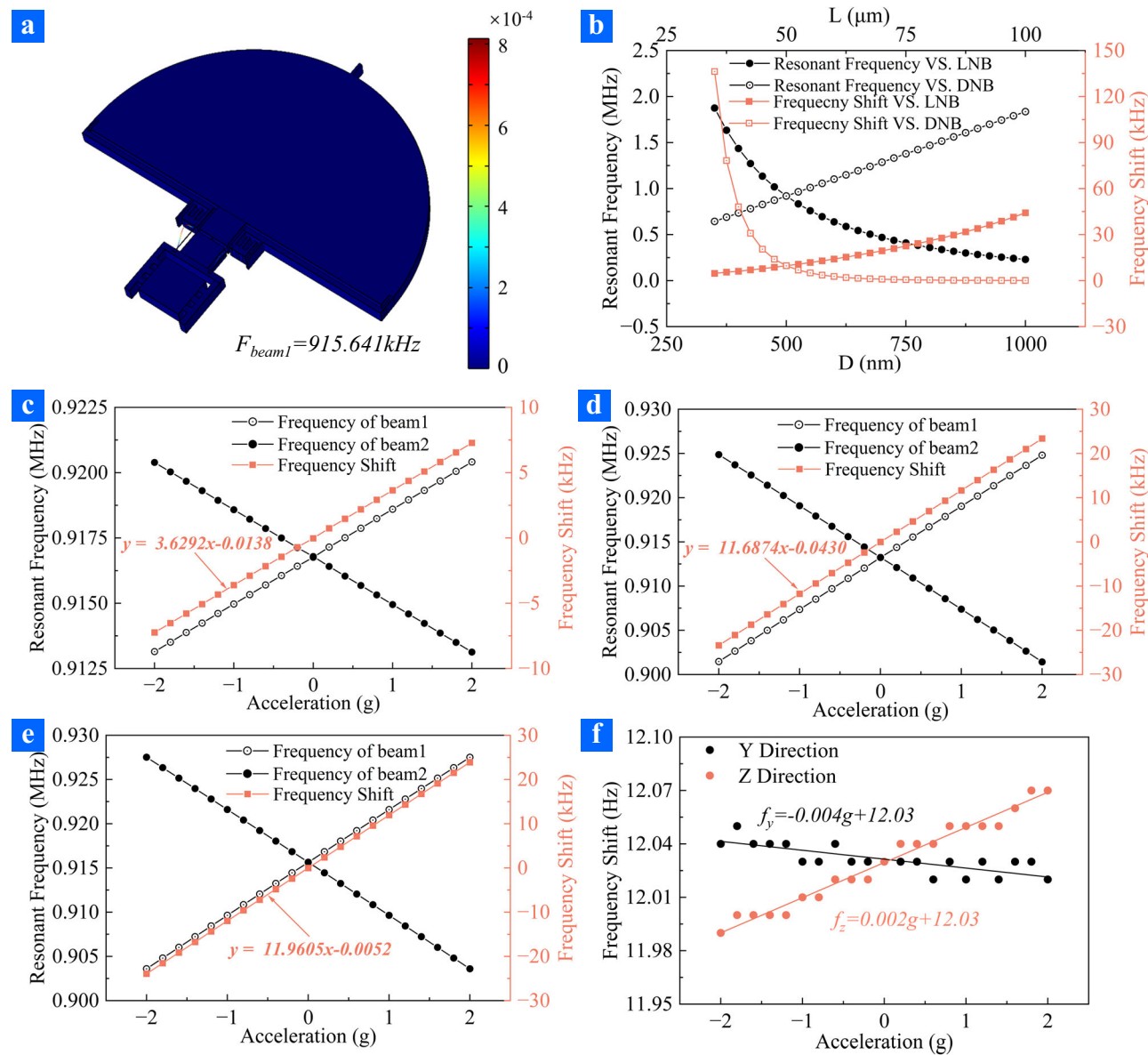

**Fig. 2 | The working mode and analysis results of the differential resonant accelerometer. a** The working mode of the differential resonant accelerometer. Resonant frequency and frequency shift of (**b**) ZnO nano-resonator with different sizes, (**c**) accelerometer without microleverage under different external accelerations, (**d**) accelerometer with microleverages under different external accelerations, (**e**) accelerometers with microleverages under different external accelerations in electronic field. **f** Resonant frequency and frequency shift of differential resonant accelerometer under different external accelerations in y-direction and z-direction.

the ZnO nano-resonator was illustrated in Fig. 2a, and the resonant frequency of the ZnO nano-resonator was determined to be 915.64 kHz as no acceleration applied on the accelerometer. When the acceleration varied from −2g to 2 g with a step of 0.2 g, the resonant frequency of the tensile ZnO nano-resonator increased linearly, while the resonant frequency of the compressive ZnO nano-resonator decreased linearly. Consequently, the differential of the resonant frequency was employed to assess the sensitivity of the resonant accelerometer. The sensitivity of the ZnO resonant nano-accelerometer without microleverages was demonstrated to be 3.629 kHz/g, while the sensitivity of the ZnO resonant nano-accelerometer with microleverages was determined to be 11.687 kHz/g. The results indicated that the microleverage was helpful to improve the sensitivity of the resonant accelerometers. Since the ZnO nano-resonator was actuated and detected electrically, the sensitivity of the accelerometer could be influenced by the electrostatic force. The sensitivity of the accelerometer was also

examined while the ZnO nano-resonator was in an electric field. As displayed in Fig. 2e, the sensitivity of the resonant accelerometer was demonstrated to be 11.961 kHz/g, while the ZnO nano-resonator was located in the air domain, and the electrical field was applied on the interface of the air and the vacuum domains, which was slightly different from it when the electrical field was absent. The accelerometer was inevitably subjected to acceleration in non-sensitivity directions, it is necessary to minimize the cross-sensitivity of the resonant accelerometer. The cross-sensitivity of the resonant accelerometer was demonstrated with FEA method, and the acceleration was varied from −2g to 2 g with step of 0.2 g in Y-direction and Z-direction, which were displayed in Fig. 2f. The frequency shifts of the symmetrically distributed ZnO nano-resonator decreased monotonously in Y-direction but increased monotonously in Z-direction. The cross-sensitivity of the accelerometer was determined to 0.0041 Hz/g in Y-direction and 0.0212 Hz/g in Z-direction, which were negligible compared to the

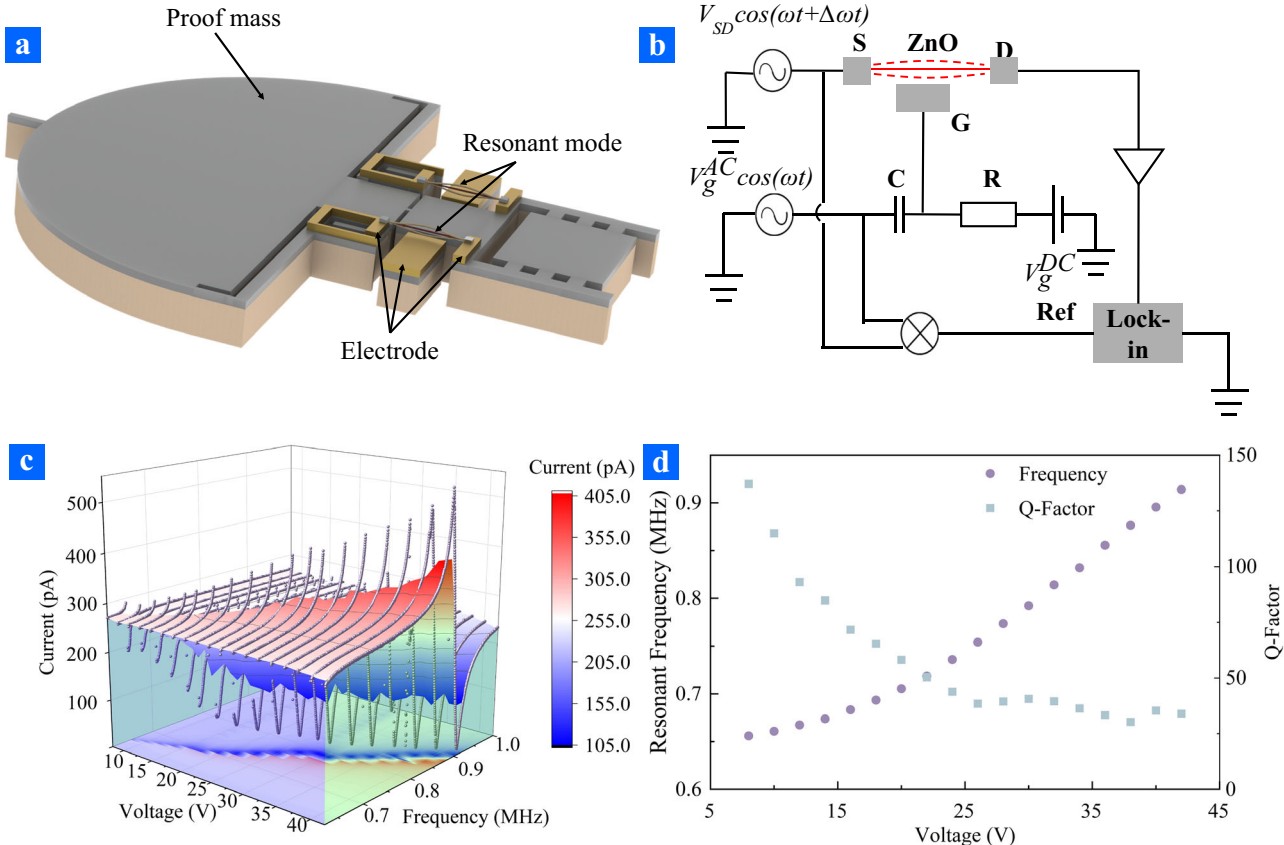

**Fig. 3 | The measurement setup and results of accelerometer resonant characterization. a** Schematic diagram of the nanowire resonant mode. **b** Circuit diagram of resonant frequency measurement circuit. **c** Current and frequency responses with different voltages applied on the gate. **d** The relationship between resonant response characterizations (resonant frequency and Q-factor) and voltages applied on the gate.

sensitivity in X-direction and would not significantly affect the sensitivity of the resonant accelerometer.

The resonant behaviors of the ZnO nano-resonator were actuated and detected electrically due to its transistor properties of semiconducting[31–33]. An AC voltage was applied on the source, and a combination of AC voltage and DC voltage was applied on the gate to actuate the ZnO nano-resonator into high frequency vibration motion. To evaluate the resonant frequency of ZnO nano-resonator, a mixing principle based on lock-in amplifier was employed to read the frequency response of the ZnO nano-resonator, as shown in Fig. 3b. The ZnO nano-resonator was placed in a vacuum environment under pressure of about $10^{-2}$ Torr to obtain high quality factor (Q factor). The current frequency curve of the ZnO nano-resonator with different DC voltage applied on the gate was displayed in Fig. 3c. It was evident that the peak current flowing through the ZnO nanowire raised with an increase of the DC voltage applied on the gate, which was attributed to the larger stretched towards the gate resulting from the DC voltage. Meanwhile, the dynamic range of the resonator was evaluated to be 19.65 dB. Additionally, the resonant frequency and the Q factor of the ZnO nano-resonator were extracted from Fig. 3c with Lorentz fittings, which were displayed in Fig. 3d. The resonant frequency increased monotonically with increasing DC voltage applied to the gate, resulting from the spring constant hardening due to the stretching of the ZnO nano-resonator. It suggests that the initial stress of the ZnO nano-resonator was small enough to work in high sensitivity detection[34–36]. More importantly, the resonant frequency tuning range of the ZnO nano-resonator was 0.65 MHz to 0.91 MHz, which may be adopted to compensate for the resonant frequency deviation caused by the fabrication error and residual stress of the accelerometer. The Q factor of ZnO nano-resonator decreased as the DC voltage applied on the gate

increased, which suggested the accelerometer to operate with small DC voltage to obtain high accuracy resonant frequency and acceleration. From the ZnO nano-resonator's open-loop measurement, the bandwidth of the ZnO nano-resonator was calculated as $\Delta f_r = \frac{f_{res}}{Q} \cong$ 4.78 ~ 26.94 kHz, which depends on the DC voltage applied on the gate.

Next, the sensitivity of the ZnO resonant nano-accelerometer was measured using a series of tilt tests. The experimental setup and measurement results were displayed in Fig. 4. To measure resonant frequency shifts under different accelerations, the device was installed on a rotary table. The acceleration from 1 g to −1 g was generated by adjusting the angle between the plate and the horizontal line with the rotary table. The resonant frequency of each ZnO nano-resonator was measured when the intersection angle was adjusted to different degrees, respectively. This enable the measurement of static acceleration sensitivity in the range of ±1 g. As a reference, right and left ZnO nano-resonators processes resonant frequency at 0.715 MHz and 0.716 MHz with no acceleration applied. The marginal resonant frequency difference in two ZnO nano-resonators could originate from the dimension difference of ZnO nanowires or the fabrication-induced residual stress. The frequency mismatch, however, can be compensated by the voltage applied on the gate.

Figure 4b displayed the typical frequency response of the ZnO nano-resonator under different acceleration, demonstrating a small variation in amplitude and signal-to-noise ratio. The resonant frequency shifts of the ZnO resonant nano-accelerometer under different accelerations were displayed in Fig. 4c. The sensitivities of the left and right ZnO nano-resonator were determined to be 8.416 kHz/g and 8.402 kHz/g, respectively. Since it is an accelerometer with a push-pull differential architecture (as displayed in Fig. 1d), the differential

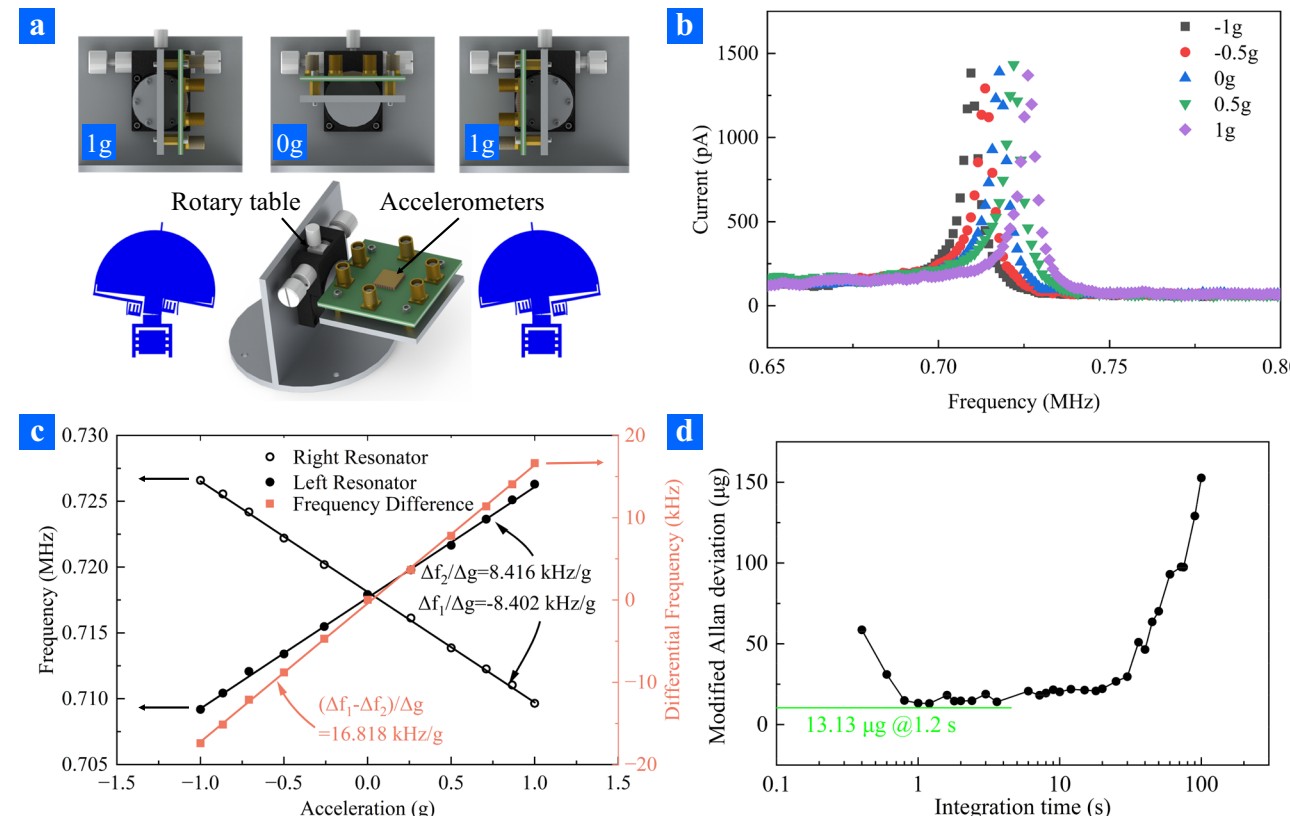

**Fig. 4 | Measurement setups and results of the ZnO resonant nano-accelerometer. a** Schematic diagram of the accelerometer measurement. **b** The typical measurement results of resonant frequency of right beam of the differential resonant accelerometer under different external acceleration. **c** The resonant frequency and differential resonant frequency of the ZnO resonant nano-accelerometer. **d** The Allan deviation of the ZnO resonant nano-accelerometer as the external acceleration was 0.

frequency sensitivity was the sum of the sensitivities of the two ZnO nano-resonators (i.e. 16.818 kHz/g). As mentioned above, the resonant frequency of the ZnO nano-resonator was 0.715 MHz, the relative frequency change of $\Delta f / f_n$ was $2.35 \times 10^4$ ppm. The non-linearity of each of ZnO nano-resonators and differential sensitivity were less than 1%. The sensitivity of the accelerometer was measured again after a month (16.298 kHz/g) demonstrating a good single device reproducibility. Additionally, over 20 accelerometers were replicated showing sensitivity over 10 kHz/g (detailed information was provided in Supplementary Materials Note S6). The consistently high sensitivity consolidates the advantages of ZnO nano-resonators.

Since the acceleration is determined from resonant frequency of ZnO nano-resonators, the frequency stability is one of the crucial aspects of a resonant accelerometer. Using the same setup shown in Fig. 3b, stability measurements of the output frequency signals was performed for 30 min with an integration time of 200 ms. The device was mounted at 0 g position, and working temperature was about 25 °C. As displayed in Fig. 4d, the Allan deviation was plotted for the device working at room temperature, showing a bias instability of 13.13 μg at 1.2 s integration time, which is consistent with the stability of the ZnO nano-resonator (detailed information was provided in Supplementary Materials Note S5).

## Discussion

As a summary, a ZnO resonant nano-accelerometer with high sensitivity was designed and prototyped based on crystalline ZnO nanowires, which possesses outstanding mechanical and electrical properties. Microleverages and notched flexure were employed to improve the sensitivity of the accelerometer, and the accelerometer structure was optimized with FEA method. For high sensitivity, a pair of

symmetrically distributed ZnO nano-resonators were built with microscope optical nanomanipulation and FIB technique. Due to the semiconducting properties of ZnO nanowires, ZnO nano-resonators were actuated and detected simultaneously, which facilitates for integration with electronic circuits. The static sensitivity of the accelerometer was measured to be 16.818 kHz/g with the bandwidth from 4.78 to 26.94 kHz, which were much larger than conventional MEMS resonant accelerometers. The Allan deviation yields a bias instability of 13.13 μg at 1.2 s integration time, while the Q factor of the accelerometer demonstrated to several hundred. The Q factor of the accelerometer is limited by the structure of the devices, which affects the resolution of the resonant accelerometer owing to the noise of the resonator. The high sensitivity of the ZnO resonant nano-accelerometer, however, could compensate the drawback of the Q factor partly. We will devote our efforts to obtain Q factor at least several thousands, which have been achieved in our previous research[37], and a compensation algorithm based on the simulation and experiment in future study to solve this deficiency effectively.

NEMS resonator based accelerometer generally have higher absolute and relative sensitivity, indicating the dimension of resonators play a key role in high precision accelerometer (detailed information was provided in Supplementary Materials Note S7). Meanwhile, the ZnO nanowire was totally oxidized owing to its wurtzite structure, which makes the ZnO resonant nano-accelerometer suitable for long term application with vacuum packaging, such as inertial navigation for robotics, unmanned aerial vehicles and wearable devices. Furthermore, such effort could encourage the combination of "top-down" methods with developed synthesized functional nanomaterials (i.e. carbon nanotubes, graphene, metal oxide nanostructures), which

would be an ideal candidate for silicon devices with performances could exceed from those.

## Methods

### Fabrication of ZnO nano-resonant accelerometer

The ZnO nanowires used to form double clamped nano-resonators were grown by CVD methods. The as-grown nanowires have a uniform morphology as seen from the SEM image in Fig. S11a, with a diameter from tens of nanometers to hundreds of nanometers and a length of tens of micrometers to hundreds of micrometers. The crystal structure and phase purity of the bulk nanowire samples were assessed using X-ray diffraction (XRD) and shown in Fig. S11b. All the relatively sharp diffraction peaks are in good agreement with the standard ZnO wurt-zite structure, a = 0.3250 nm and c = 0.5206 nm[38]. Transmission elec-tron microscope (TEM) analysis was carried out to confirm the crystalline structure of ZnO nanowires. The results show that the ZnO nanowires grow in the [0001] direction with a wurtzite structure, consistent with XRD results[39].

The double clamped ZnO nano-resonator was fabricated by optical microscopy nanomanipulation technique and FIB technique, as dis-played in Fig. S12. The synthesized ZnO nanowires were ultrasonically dispersed in ethanol for several minutes to separate and disperse the individual ZnO nanowire from the ZnO nanowires cluster. A drop of the solvent containing ZnO nanowires was dispersed onto the grid, and then a tungsten needle was used to pick up a single ZnO nanowire and place it on the accelerometer with predesigned structures. To clamp ZnO nanowires suspended across the source and drain, an FIB tech-nique (16 keV Ga+ with a 25 pA current aperture) was carried out to deposit Pt onto the interface between the nanowire and electrodes.

### FEA methods

The FEA for the accelerometer was performed with COMSOL Mluti-physics. The resonant mode shape and performance of the ZnO nanobeam was analyzed by parametric sweep the diameter and length of the ZnO nanobeam, and the simulation model was displayed in Fig. S13, where mass density (5670 kg/m$^3$), Poisson ratio (0.3), and Young's modulus (150 GPa) were employed as material parameters. The optimization model of the notched flexure was displayed in Fig. S14, as boundary loads and fixed constraint were applied on the boundary perpendicular to the ZnO nanobeam. The optimization model for the microleverage structure was displayed in Fig. S15, as the distance between the pivot beam and output beam increased from 6 to 120 μm without other changes and sweep the width of the support beam from 1 μm to 10 μm with a step of 1 μm and keep other para-meters invariant. The simulation model for the accelerometer sensi-tivity force was displayed in Fig. S16, as the acceleration was varied from −2 g to 2 g with a step of 0.2 g applied on the structure. The resonant frequency shifts caused by thermal stress was analyzed by sweep working temperature from 25 to 50 °C with a step of 5 °C (Fig. S17), where the coefficient of thermal expansion (CTE) of ZnO nanowires and silicon were sought from D. Taylor et al. Transactions and Journal of the British Ceramic Society, v83, No. 1, p1, R.R. Rebber et al. Journal of Applied Physics, v41, No. 13, and C.A. Swenson et al., Journal of Physical and Chemical Reference Data, v12, No. 2.

## Data availability

The associated data generated in this study have been deposited in the figshare database under accession code https://doi.org/10.6084/m9.figshare.24716853.

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

## Acknowledgements

This research was supported by National Key R&D Program of China, (Grant Number: 2018YFA0703200, D.W.), National Natural Science Foundation of China (Grant Number: 52103224, T.W. and Grant Number: 62074138, T.R.), the Fundamental Research Funds for the Central Universities (Grant Number: DUT22LAB405, D.W.), Liaoning Huanghai Laboratory and Ningbo Institute of Dalian University of Technology. The authors gratefully acknowledge the assistance of researchers of the transmission electron microscopy during the experiments. Special thanks are due to Mrs. Xiaoxia Gao in Instrumental Analysis Center of Dalian University of Technology for the assistance with TEM analysis.

## Author contributions

D.W. responsible for the funding and resources acquisition, super-vising the project, revising the manuscript. P.X. responsible for most of the investigations, methodology development, data collection/analysis, writing and editing the manuscript. J.H. and T.W. helped to take most of the experimental tests and editing the manuscript. Y.C. T.R. and L.L. helped the XRD image. Y.L. and X.C. helped to take the experimental pictures. C.L., Y.C. and L.S. completed the TEM image.

## Competing interests

The authors declare no competing interests.
