## [Peer Review File · Nature Communications]

REVIEWER COMMENTS

Reviewer #1 (Remarks to the Author):

The manuscript titled "A ZnO nano-resonant accelerometer with ultra-high sensitivity" has been written well.

The results of the manuscript are pretty good and the manuscript contains fairly enough data for possible publication however, it will be better if the authors would address the following quires,

1. What is the reproducibility of the sensitivity of accelerometers?
2. All devices' performance should be given in the form of a statistical report with a chart
3. Language is poor in several places. Hence, the authors should recheck the manuscript.
4. In several places, the authors did not mention the full names of words like SOA, SOI, SOG. Please check whole manuscript.
5. According to the authors statement, "To enhance the sensitivity of the accelerometer, a microleverage mechanism was implemented",. To support this, authors need to provide the sensitivity of without microleverage mechanisms for a better understanding of sensitive improvements.
6. On what basis the distance of 36 um was chosen?
7. What is the use of 0.3 um of Gold? Have tried other thicknesses of Au?
8. How the authors confirmed the single crystal ZnO nanowires? Supporting data will be required.
9. Quality of the several figures needs to be improved. Several letters are not visible.

Reviewer #2 (Remarks to the Author):

The paper presents a novel resonant accelerometer structure where the boundary of a nanowire resonator is coupled to a proof-mass, inducing shift in frequency upon acceleration. The experimental results are conclusive and well prepared. The fabrication process has successfully implemented the targeted concept. Overall, the paper can be published at Nature Comm. once addressing the followings:

1) Resonant accelerometer may offer some advantages from static noise perspective. However, the frequency is not only sensitive to acceleration, but is typically highly correlated with operation temperature and pressure. Suppressing cross-axis sensitivity between temperature and pressure, and acceleration is very challenging. Authors have claimed, very briefly, that all other physical sensitivities (except acceleration) are suppressed by differential nature of their device. This is a major claim and must be elaborated and justified. For example, it should be shown the resonance frequencies have exactly the same temperature characteristics regardless of acceleration direction. This may not be the case, considering nonuniform nonlinearity in ZNO nanowires that may affect their temperature coefficient differently.

2) The oscillators that is created around resonant accelerometer suffer from various instability processes such as interface circuit noise (i.e., frequency and phase white and flicker noise), as well as resonator aging, temperature instability and nonlinearity. Therefore, despite the apparent advantages of resonant accelerometer over static counterpart, numerous new challenges again generation of stable oscillation, consumption of excessive power to sustain oscillations, and finally limitations added by analog to digital conversion for frequency-shift read-out is added at the system-level. Therefore, static accelerometers have been the superior choice and provide significantly higher system-level performance. Authors must provide data evaluating the stability of oscillations. Allan Deviation is the best metric for this purpose.

3) The resolution of the resonant accelerometer is affected by phase noise of oscillator, which depends on quality factor (Q). The Q of presented resonance modes are very low, particularly compared to silicon-based resonant accelerometers that provide Qs in excess of millions. How does the presented work address this deficiency?

4) Sensitivity is one of performance metrics for evaluation of accelerometer. The primary metrics are dynamic range and bandwidth. None of these are discussed / no data is presented.

5) ZNO is a moderate piezoelectric material. I appreciate the nanowire fabrication aspect of the presented work. However, AlScN and HZO films can also be deposited with few nanometers thickness and patterned by electron-beam lithography to the same size of nanowires, while offering significantly higher piezoelectric coefficient. Please justify the advantage of ZNO nanowire over alternative piezoelectric films with larger coupling.

Reviewer #3 (Remarks to the Author):

I thank the authors for their submission. I have, however, several comments about it. Overall, though the concept is somewhat interesting, the research status is so preliminary and naive that the paper cannot be accepted in this form and several modifications are mandatory.

General comments

1. Grammar, use of verb tenses, emphasis of sentences, all need a proof revision by a native or fluent English speaker.
2. At the basis of all authors claims is the use of ZnO for NEMS resonators as transduction mean, and the claim of the highest sensitivity in the literature. This is not true. Other works already used NEMS resonators, and reached larger sensitivity than the authors. E.g. Miani in proc. IEEE MEMS 2021 and later in proc. IEEE Inertial 2022 reached scale factors as large as 1.4 MHz/g which is two orders of magnitude larger than the present work. Authors need for sure to mention those works.
3. Additionally, those results were obtained with monolithic Silicon structures (also NEMS gauges were made of Si, with 250 nm width), with no need of extra steps and extra materials. Authors should mention, indeed, the drawbacks of mixing different materials with different CTE and TCE in terms of overall system stability.
4. Characterization is absolutely preliminar. Accelerometers have been on the market for 30 years, and just showing a rough measure of the scale factor with only a few points is definitely not enough. Any modern paper on accelerometers shall show (i) scale factor up to the full-scale range and corresponding nonlinearity, (ii) noise in terms of PSD or even better in terms of root Allan variance and possibly also (iii) temperature and vibration stability. It is only the combination of these parameters that tells you whether an accelerometer is suited or not for specific applications. Just boosting the scale factor, e.g., might simply lower the full-scale range and make the device inadequate for several applications.

Minor comments

5. Please complete the literature review with the works mentioned above and other recent works showing performance much better than those presented here.

6. No mention is given to the packaging, which instead has a major role in state-of-the-art performance of accelerometers, especially in terms of stability vs temperature, bending ecc...

7. The claim that "NEMS accelerometers are extensively used" is untrue. Most of the published accelerometers are MEMS based, while NEMS based are only a few.

8. Once again, the increase in sensitivity is not the fundamental parameter for high precision inertial sensors, as long as it is not accompanied by other improvements (noise, wide FSR).

9. The claim that resonant accelerometers are not vulnerable to electronic noise is wrong. Phase noise arises from the electronic circuit and affects the performance. Unfortunately, authors were not able to predict the noise limits and no noise measurements were shown in the manuscript.

10. Which is the target application for your device? Authors mention in a generic way next generation devices, but it is impossible to conceive a device for all applications! Device design shall be indeed application driven, so authors shall specify the target application and related aforementioned parameters (FSR, noise, linearity, bandwidth...).

11. Why are sensitivity predictions made only via FEA? There is a rather simple mechanical calculation to predict the scale factor...

12. ZnO material parameters (CTE, TCE ecc...) shall be provided and their impact on a silicon-based structure with different parameters, when temperature changes, shall be given.

13. The sole experimental measurements that are shown are rather poor: 5 points only to estimate the scalefactor, some of which are clearly out of the fitting line is not a good demonstration of a "high-precision" device. Instead, it seems very poor.

Comments from Reviewer #1:

Comment 1:

What is the reproducibility of the sensitivity of accelerometers?

Answers:

✓ We appreciate the insightful and valuable comment from the reviewer. In this revised version, the discussion about the sensitivity reproducibility of the ZnO resonant nano-accelerometers was added. The sensitivity of the ZnO resonant accelerometers was reproducible. One device's performance was presented in the paper, however, the performance of the device was checked many times in our experiment. The longest interval time of the experiment was over a month, and the scale factor obtained with least square method was demonstrated to have a good reproducibility. Additionally, over 20 accelerometers were prepared with our technique, and the sensitivity of accelerometers was over 10 kHz/g for majority of them, which benefited from nanoscale ZnO resonators.

✓ The detailed explanation was added in this revision on page 2:

The sensitivity of the accelerometer was measured again after a month (16.298 kHz/g) demonstrating a good single device reproducibility. Additionally, over 20 accelerometers were replicated showing sensitivity over 10 kHz/g. (Detailed information was provided in supplementary materials Note S6) The consistently high sensitivity consolidates the advantages of ZnO nano-resonators.

✓ Another detailed explanation was added in supplementary materials Note S6:

Note S6 Sensitivity reproducibility of ZnO resonant nano-accelerometers

The sensitivity of the accelerometer was checked many times, and the longest interval time was over a month. The sensitivity was measured to be 16.298 kHz/g, demonstrating a good single device reproducibility. In addition, over 20 ZnO resonant nano-accelerometers were prepared by optical microscope manipulation and FIB technology. The sensitivity of the devices was measured as introduced in the manuscript, majority of them were up to 10 kHz/g, which benefited from the ZnO nano-resonators.

Figure S7 (a) The sensitivity of ZnO resonant nano-accelerometer after a month. (b) Sensitivity statistical diagram of ZnO resonant nano-accelerometers

Comment 2:

All devices' performance should be given in the form of a statistical report with a chart.

Answer:

✓ We appreciate the insightful and valuable comment from the reviewer. In this revised version, for the better understandings of readers, a scatter plot was offered to highlight the different performance of the resent resonant accelerometers. It will be easier for readers to understand and compare the performance of different resonant accelerometers. Additionally, we have

included the performance of other accelerometers based on NEMS resonators to illustrate the current state of development of resonant accelerometers.

- ✓ The detailed explanation was added in this revision on page 13:

NEMS resonator based accelerometer generally have higher absolute and relative sensitivity, indicating the dimension of resonators play a key role in high precision accelerometer. (Detailed information was provided in supplementary materials Note S7).

- ✓ Another detailed explanation was added in supplementary materials Note S7:

Note S7 The comparison of our device and other reported resonant accelerometers

Figure S10 The sensitivity and resonant frequency of this work and other resonant accelerometers.

Table S1. Comparison of the resonant frequency, sensitivity and relative sensitivity of previous devices and our device

Number	Reference	Sensitivity (kHz)	Resonant frequency (kHz)	relative sensitivity (ppm)
1	Xie et al.	0.2975	16.061	18520
2	Yang et al.	0.052	27	1930
3	Comi et al.	0.455	58	7840
4	Caspani et al.	0.25	84	2980
5	Wang et al.	1.1533	135	8540
6	Seshia et al.	0.017	173	98.2659
7	Gabriele et al.	0.0181	332	54.5181
8	Zhao et al.	2.752	352.2	7810
9	Aikele et al.	0.07	400	175
10	Zou et al.	5.61	420	13360
11	Pinto et al.	0.022	459	47.9303
12	Maro et al.	1400	14040	97000
	This paper	16810	716	23480

✓ The relevant references are listed as follows:

1 T. Miani, *et al.*, Resonant accelerometers based on nanomechanical piezoresistive transduction, 34th IEEE MEMS, Electr Network, 192-195, (2021)

2 Comi, C. et al. A Resonant Microaccelerometer With High Sensitivity Operating in an Oscillating Circuit. *Journal of Microelectromechanical Systems* 19, 1140-1152, (2010).

3 Caspani, A., Comi, C., Corigliano, A., Langfelder, G. & Tocchio, A. Compact biaxial micromachined resonant accelerometer. *Journal of Micromechanics and Microengineering* 23, 1-11, (2013).

- 4 Aikelea, M. et al. Resonant accelerometer with self-test. *Sensors and Actuators A* 92, 161-167, (2001).
- 5 Seshia, A. A. et al. A vacuum packaged surface micromachined resonant accelerometer. *Journal of Microelectromechanical Systems* 11, 784-793, (2002).
- 6 Pinto, D. et al. A Small and High Sensitivity Resonant Accelerometer. *Procedia Chemistry* 1, 536-539, (2009).
- 7 Vigevani, G., Goericke, F. T., Pisano, A. P., Izyumin, I. I. & Boser, B. E. in 2012 IEEE International Frequency Control Symposium Proceedings.
- 8 Yang, B., Zhao, H., Dai, B. & Liu, X. A new silicon biaxial decoupled resonant micro-accelerometer. *Microsystem Technologies* 21, 109-115, (2014).
- 9 Wang, S., Wei, X., Zhao, Y., Jiang, Z. & Shen, Y. A MEMS resonant accelerometer for low-frequency vibration detection. *Sensors and Actuators A: Physical* 283, 151-158, (2018).
- 10 Pandit, M. S. et al. An Ultra-High Resolution Resonant MEMS Accelerometer. 2019 IEEE 32nd International Conference on Micro Electro Mechanical Systems (MEMS), 664-667, (2019).
- 11 Zhao, C. et al. A Resonant MEMS Accelerometer With 56 ng Bias Stability and 98ng/Hz^{1/2} Noise Floor. *Journal of Microelectromechanical Systems* 28, 324-326, (2019).
- 12 Ding, H., Wu, C. & Xie, J. A MEMS Resonant Accelerometer With High Relative Sensitivity Based on Sensing Scheme of Electrostatically Induced Stiffness Perturbation. *Journal of Microelectromechanical Systems* 30, 32-41, (2021).

Comment 3:

Language is poor in several places. Hence, the authors should recheck the manuscript.

Answers:

- ✓ We appreciate the insightful and valuable comment from the reviewer. In this revised version, we have conducted a thorough language review and modification of the manuscript to ensure the language smooth and accurate. To further enhance the linguistic quality of the article, we invited Dr. Tiesheng Wang to polish the manuscripts, facilitating readers' comprehension and understanding of the content. All changed places were highlighted.

Comment 4:

In several places, the authors did not mention the full names of words like SOA, SOI, SOG. Please check whole manuscript.

Answers:

- ✓ We appreciate the insightful and valuable comment from the reviewer. In this revised version, we have rechecked the entire manuscript and ensured that the full names were provided when these abbreviations were introduced for the first time. Subsequently, these abbreviations will be used in the manuscript to maintain conciseness. We thank you again for your meticulous review and suggestions, and we will ensure that all abbreviations are clearly defined in this version.

Comment 5:

According to the authors statement, “To enhance the sensitivity of the accelerometer, a microleverage mechanism was implemented”. To support this, authors need to provide the

sensitivity of without microleverage mechanisms for a better understanding of sensitive improvements.

Answers:

✓ We appreciate the insightful and valuable comment from the reviewer. In this revised version, the schematic diagrams and mechanisms of different microleverage structures were presented in supplementary materials. The microleverage structure was selected based on the need of the accelerometer, enhancing the performance of accelerometer and facilitating electrodes' fabrication. To further enhance the performance of the accelerometer, COMSOL Multiphysics was adopted to optimize the parameters of the microleverage structure. Additionally, as the reviewer's suggestions, we have introduced a comparison between the sensitivity of accelerometers without microleverage structures and those with a microleverage structure. This addition serves to effectively illustrate the sensitivity improvement attributable to the incorporation of the microleverage structure.

✓ The detailed explanation was added in this revision on page 4:

To enhance the sensitivity of the accelerometer, a microleverage mechanism was implemented. According to the position of the pivot beam, input beam, and output beam of the microleverage, the microleverage was classified into three types. Herein, the first type of microleverages mechanism was adopted to improve the sensitivity of the accelerometer and facilitate electrodes fabrication. The structure of the miceoleverage was optimized with FEA method which evaluated its performance by the deformation of the ZnO nanobeam. (Detailed information was provided in supplementary materials Note S3) Herein, the support beam of the microleverage mechanism was determined to be 3 μm with a distance of 36 μm from the output beam.

- ✓ Another detailed explanation was added in this revision on page 8~9:

The sensitivity of the accelerometer was analyzed with FEA method, and the results were displayed in Figure 2. The working mode of the ZnO nano-resonator was illustrated in Figure 2a, and the resonant frequency of the ZnO nanobeam was determined to be 915.64 kHz as no acceleration applied on the accelerometer. When the acceleration varied from -2g to 2g with a step of 0.2g, the resonant frequency of the stretched ZnO nano-resonator increased linearly, while the resonance frequency of the compressed ZnO nano-resonator decreased linearly. Consequently, the differential of the resonant frequency was utilized to assess the sensitivity of the accelerometer, and the accelerometer without microleverages was demonstrated to be 3.629 kHz/g while the accelerometer with microleverages was determined to be 11.674 kHz/g. The result indicated that the microleverage was helpful to improve the sensitivity of the resonant accelerometers.

Figure 2 The working mode and analysis results of the differential resonant accelerometer. (a) The working mode of the differential resonant accelerometer. Resonant frequency and frequency shift of (b) ZnO nano-resonator with different sizes, (c) accelerometer without microleverage under different external accelerations, (d) accelerometer with microleverages under different external accelerations, (e) accelerometers with microleverages under different external accelerations with electronic force. (f) Resonant frequency and frequency shift of differential resonant accelerometer under different external accelerations in in y-direction and z-direction.

Comment 6:

On what basis the distance of 36 μm was chosen?

Answers:

- ✓ We appreciate the insightful and valuable comment from the reviewer. In this revised version, the distance of pivot beam was analyzed using COMSOL Multiphysics. The microleverage structure was demonstrated to efficiently amplify the force acting on the input beam. Nonetheless, the amplification factor of the microleverage structure was contingent upon the position of the pivot beam. To enhance the effective amplification of the force on the input beam, COMSOL Multiphysics was employed for parameter optimization. A parametric sweep was then utilized to analyze the position of the pivot beam by calculating the deformation of the ZnO nano-resonator. The results revealed that the deformation of the ZnO nanobeam was maximized when the pivot beam was positioned 36 μm away from the output beam.
- ✓ The detailed explanation was added in this revision on page 4:

To enhance the sensitivity of the accelerometer, a microleverage mechanism was implemented. According to the position of the pivot beam, input beam, and output beam of the microleverage, the microleverage was classified into three types. Herein, the first type of microleverages mechanism was adopted to improve the sensitivity of the accelerometer and facilitate electrodes fabrication. The structure of the miceoleverage was optimized with FEA method which evaluated its performance by the deformation of the ZnO nanobeam. (Detailed information was provided in supplementary materials) Hence, the support beam of the microleverage mechanism was determined to be 3 μm with a distance of 36 μm from the output beam.

- ✓ Another detailed explanation was added in this revision in supplementary materials Note S3:

Note S3 Design of the microleverage

To enhance the sensitivity of the accelerometer, a microleverage mechanism was implemented. According to the position of the pivot beam, input beam, and output beam of the microleverage, the microleverage was classified into three types. The first type of microleverage has a flexible support beam between the input beam and the output beam. When the dynamic arm is greater than the resistance arm, this structure can simultaneously amplify inertial forces and micro-displacements, and also has the function of changing the direction of inertial forces and micro-displacements. The second type microleverage mechanism has input and output beams on the same side of the support beam, with the output beam closer to the support beam. This type of structure is mainly used to amplify inertial forces. The third type microleverage mechanism has input and output beams on the same side of the support beam, with the input beam closer to the support beam. This type of structure is mainly used to amplify micro-displacements. Herein, the first type of microleverages mechanism was adopted to improve the sensitivity of the accelerometer and facilitate electrodes fabrication. The structure of the miceoleverage was optimized with FEA method which evaluated its performance by the deformation of the ZnO nanobeam. The results were displayed in Figure S4d and Figure S4e. The deformation of the ZnO nanobeam initially increased and then decreased as the distance between the support beam and output beam increased from 6 to 120 μm , with the maximum deformation occurring when the distance was 36 μm , as shown in Figure S4d. As shown in Figure S4e, it is evident that the narrower the width of the support beam, the larger the deformation of the ZnO nanobeam. However, if the width of the support was too small, the stability of the microleverage may be compromised.

Hence, the support beam of the microleverage mechanism was determined to be 3 μm with a distance of 36 μm from the output beam.

Figure S4. The microleverage structure of accelerometer (a) the first type microleverage structure, (b) the second type microleverage structure, (c) the third type microleverage structure, (d) the optimization of the microleverage support beam position, (e) the optimization of the microleverage support beam width.

Comment 7:

What is the use of 0.3 μm of Gold? Have tried other thicknesses of Au?

Answers:

- ✓ We appreciate the insightful and valuable comments from the reviewer. The 0.3 μm gold layer was used as an electrode to drive and detect the ZnO nano-resonators, which were employed to evaluate the acceleration with the resonant frequency shift. In the fabrication process of other ZnO resonator devices, 0.2 μm gold has also been used. Experiments and literatures

review have shown that gold layers with a thickness of 0.05~0.3 μm can effectively drive and detect the resonant response of nano-resonator.

Comment 8:

How the authors confirmed the single crystal ZnO nanowires? Supporting data will be required.

Answers:

- ✓ We appreciate the insightful and valuable comment from the reviewer. In this revised version, the crystalline structure of ZnO nanowires was analyzed with X-Ray Diffraction (XRD) and transmission electron microscope (TEM). Due to the semiconducting properties of semiconducting and excellent mechanical properties, ZnO nanowires were employed as nano-resonators to measure acceleration with resonant frequency shifts. Hence, it was unnecessary to require ZnO nanowires to be single crystal. In this revision, for accuracy and normal statement, *single crystal ZnO nanowires* were replaced by *crystalline ZnO nanowires*. Meanwhile, the XRD pattern was employed to analyze the crystalline structure of ZnO nanowires. The results indicate that the ZnO nanowires have a standard wurtzite structure. To further elucidate the crystal structure of the ZnO nanowires, the TEM analysis was carried out to confirm the crystalline structure of ZnO nanowires. The results show that the ZnO nanowires grow in the [0001] direction with a wurtzite structure, consistent with the XRD results.
- ✓ The detailed explanation was added in this revision on page 13:

The ZnO nanowires used to form double clamped nanobeam structure were grown by CVD methods. The as-grown nanowires have a uniform morphology as seen from the SEM image in Figure S10(a), with a diameter from tens of nanometers to hundreds of nanometers and a

length of tens of micrometers to hundreds of micrometers. The crystal structure and phase purity of the bulk nanowire samples were assessed using X-ray diffraction (XRD) and shown in Figure S10(b). All the relatively sharp diffraction peaks are in good agreement with the standard ZnO wurtzite structure, $a = 0.3250$ nm and $c = 0.5206$ nm³⁸. Transmission electron microscope (TEM) analysis was carried out to confirm the crystalline structure of ZnO nanowires. The results show that the ZnO nanowires grow in the [0001] direction with a wurtzite structure, consistent with the XRD results³⁹.

✓ Another detailed explanation was added in this revision in supplementary materials Note S8:

Note S8 Characterization of ZnO nanowires

Figure S10. The SEM image (a) and X-ray diffraction (XRD) patterns (b) of ZnO nanowires (c) low magnification TEM image of ZnO nanowire, (d) the electron diffraction (SAED) pattern of ZnO nanowire at the white circle in (c), (e) HRTEM image of ZnO nanowire at the white square in (c)

Comment 9:

Quality of the several figures needs to be improved. Several letters are not visible.

Answers:

- ✓ We appreciate the insightful and valuable comment from the reviewer. We have rechecked all the images and optimized those that were unclear, ensuring that all graphics have high clarity and readability. Specifically, the few unclear letters you mentioned have been corrected. We will ensure that high-quality images are provided in the final version for readers to better understand our research content.

Comments from Reviewer #2:

Comment 1:

Resonant accelerometer may offer some advantages from static noise perspective. However, the frequency is not only sensitive to acceleration, but is typically highly correlated with operation temperature and pressure. Suppressing cross-axis sensitivity between temperature and pressure, and acceleration is very challenging. Authors have claimed, very briefly, that all other physical sensitivities (except acceleration) are suppressed by differential nature of their device. This is a major claim and must be elaborated and justified. For example, it should be shown the resonance frequencies have exactly the same temperature characteristics regardless of acceleration direction. This may not be the case, considering nonuniform nonlinearity in ZNO nanowires that may affect their temperature coefficient differently.

Answers:

- ✓ We appreciate the insightful and valuable comments from the reviewer. In this revised version, the expression about the differential feature has been changed to improve its rigorosity and accuracy. Additionally, finite element analysis results have been incorporated to substantiate the presented expression. The differential resonant accelerometer has some advantages in static noise, and can suppress the majority of the resonant frequency shifts caused by temperature, pressure and other factors. It is important to note, however, that complete consistency in the size and properties of the two ZnO nanowires cannot be guaranteed. Consequently, resonant frequency fluctuations caused by temperature, pressure, and other factors may not be entirely eliminated. Therefore, a single-point anchor with unloading grooves was adopted to facilitate releasing stress coupled to ZnO nano-resonator.

- ✓ The detailed explanation was added in this revision on page 3:

As a foundation for its high performance, a pair of ZnO nanowires were fixed on the proof mass which swung around a notched flexure when subjected to external acceleration. This leads to a change in axial stress in the symmetrically distributed ZnO nanowires resulting in a shift in the resonant frequency. Compared to the single resonator design, the prototyped device evaluated the acceleration with a differential architecture, which could eliminate the majority of the resonant frequency shifts caused by temperature changes intrinsically.

- ✓ Another detailed explanation was added in this revision on page 3:

A single-point anchor with unloading grooves was adopted to reduce external stress coupled to the ZnO nano-resonator. (Detailed information was provided in supplementary materials Note S4)

- ✓ Another detailed explanation was added in supplementary materials Note S4:

Note S4 Design of the unloading grooves anchor

The differential resonant accelerometer can suppress the majority of the resonant frequency shifts caused by temperature, pressure and other factors. The size and properties of the two ZnO nanowires, however, cannot guaranteed to be completely consistent. So the resonant frequency fluctuations caused by temperature, pressure and other factors cannot eliminate completely. In the configuration involving a silicon structure, a ZnO nanowire, and a metal electrode in direct contact. Consequently, owing to distinct thermal expansion coefficients among these three materials, internal stresses are introduced as a consequence of thermal expansion. Specifically, axial stress induced by thermal expansion directly impacts the resonant frequency of the ZnO nano-resonator. To effectively address this phenomenon, a differential design of device structures and parameter optimization could eliminate the

majority of the frequency shifts caused by structural expansions and internal stresses. Furthermore, the design of an anchor structure that releases stress promptly can also alleviate the impact of thermal stress on the ZnO nano-resonator. Figure S5 illustrates schematic diagrams depicting rectangular anchor points and unloading anchor points, respectively.

Figure S5. The anchor structure of accelerometer. (a) The rectangle anchor structure. (b) The unloading anchor structure. The resonant frequency shifts of the rectangle anchor structure (c) and the unloading anchor structure (d).

To investigate the influence of thermal stress on the three anchor points, the resonant frequency shifts of the ZnO resonant nano-accelerometer were analyzed using Finite Element Analysis. The operational temperature ranged from 25°C to 50°C. The Young's modulus values for ZnO and Si were assumed to be 150 GPa and 120 GPa, respectively. The coefficients of thermal expansion for ZnO and Si were obtained from sources such as D. Taylor *et al. Transactions and Journal of the British Ceramic Society*, v83, No. 1, p1, R.R.

Rebber *et al. Journal of Applied Physics*, v41, No. 13, and C.A. Swenson *et. al, Journal of Physical and Chemical Reference Data*, v12, No. 2, within the specified temperature range. As illustrated in Figure S5, the differentially designed architecture successfully mitigated the majority of frequency shifts attributed to temperature variations. Furthermore, the structure with unloading grooves contributed to the stability improvement by facilitating the release of stress resulting from temperature changes.

Comment 2:

The oscillators that is created around resonant accelerometer suffer from various instability processes such as interface circuit noise (i.e., frequency and phase white and flicker noise), as well as resonator aging, temperature instability and nonlinearity. Therefore, despite the apparent advantages of resonant accelerometer over static counterpart, numerous new challenges again generation of stable oscillation, consumption of excessive power to sustain oscillations, and finally limitations added by analog to digital conversion for frequency-shift read-out is added at the system-level. Therefore, static accelerometers have been the superior choice and provide significantly higher system-level performance. Authors must provide data evaluating the stability of oscillations. Allan Deviation is the best metric for this purpose.

Answers:

- ✓ We appreciate the insightful and valuable comments from the reviewer. In this revised version, a feedback loop was prepared to trace the resonant frequency fluctuation of the accelerometers. As the reviewer mentioned, the resonant frequency of the resonators was affected by the various instability factors such as interface circuit noise, resonator aging, temperature instability and nonlinearity. To evaluate the stability of the ZnO resonant nano-accelerometer,

the device was mounted at 0g position, and the working temperature was about 25 °C. The resonant frequency differential was measured with the feedback loop.

- ✓ The detailed explanation was added in this revision on page 11:

Since the acceleration is determined from resonant frequency of ZnO nano-resonators, the frequency stability is one of the crucial aspects of a resonant accelerometer. Using the same setup shown in Figure 3(a), stability measurements of the output frequency signals was performed for 30 minutes at an integration time of 200 ms. The device was mounted at 0g position, and working temperature was about 25 °C. As displayed in Figure 4d, the Allan deviation was plotted for the device working at room temperature, showing a bias instability of 13.13 ug at 1.2 s integration time. (Detailed information was provided in supplementary materials Note S5)

- ✓ Another detailed explanation was added in this revision in supplementary materials Note S5:

Note S5 Schematic of the feedback loop measurement

Figure S6. Schematic of the response of the mixing current as a function of driving frequency

The feedback loop was prepared to trace the resonant frequency of the accelerometers with a lock-in amplifier. As displayed in Figure 3b, an AC voltage was applied on the source, and a combination of AC voltage and DC voltage was applied on the gate to actuate the ZnO nano-resonator into high frequency vibration motion. Meanwhile, a mixing principle based on lock-in amplifier was employed to read the frequency response of the ZnO nano-resonator. To trace the resonant frequency, the current value was employed, keeping I around a reference value I_{ref} by varying f . As displayed in Figure S6, when I remains between I_{min} and I_{max} , the driving frequency was not changed and, accordingly, the shift of the resonant frequency was calculated from I and the slope of the frequency around I_{ref} . The feedback time can be made as low as 50 ms. The feedback is interrupted repeatedly (typically every 600 s) during ~ 10 s for a control of the lineshape of the current-frequency (I as a function of f): if the resonant lineshape significantly differs from that measured previous, the recorded data were discarded.

Here, we indicated typical parameters used in the feedback loop. We often set the reference current to $I_0 + 0.5(I_1 - I_0)$ on the low frequency part of the current-frequency curve, I_1 being the highest current of the curve; but I_{ref} can also be lowered down to $I_0 + 0.3(I_1 - I_0)$ where the slope of the curve was steeper. We set the thresholds I_{max} and I_{min} so that $(I_{\text{max}} - I_{\text{min}}) / I_0$ is typically 0.5 (we also tried values between 0.1 and 0.7). The current range ($I_{\text{max}} - I_{\text{min}}$) corresponds to a frequency range that is typically from 1 kHz to 10 kHz, which depends on the bandwidth of the resonators. For the bias instability measurement, the feedback time was set as 200 ms. The ZnO resonant nano-accelerometer was mounted at 0g position, and working temperature was about 25 °C. The resonant frequency differential trace as function of time was displayed in Figure S7.

Figure S7 The resonant frequency differential trace as function of time

Comment 3:

The resolution of the resonant accelerometer is affected by phase noise of oscillator, which depends on quality factor (Q). The Q of presented resonance modes are very low, particularly compared to silicon-based resonant accelerometers that provide Qs in excess of millions. How does the presented work address this deficiency?

Answers:

- ✓ We appreciate the insightful and valuable comments from the reviewer. The resolution of resonant accelerometer was related with the phase noise of oscillator, which depended on the quality factor (Q). And the Q of the presented resonators were low, which was limited by the structure of the resonator and the working temperature. The resolution of the accelerometer, however, depended on the stability ($\frac{\Delta f}{f}$) and sensitivity of the accelerometers. Benefited from

the ZnO nano-resonator, the sensitivity of the accelerometer was up to 16.818 kHz/g, which was a compensation for the resolution of the accelerometer. In order to suppress the frequency fluctuation, a single-point anchor with unloading grooves was adopted to reduce the external stress coupled to the ZnO nano-resonator. In addition, the Q of ZnO nano-resonators was demonstrated to be about several thousands in previous works, which could be worked out to improve the resolution of the resonant accelerometer. In future study, we will make effort to build the compensation algorithm based on the simulation and experiment, which indicated that a linear relationship between the frequency shifts and the temperature.

- ✓ The detailed explanation was added in this revision on page 3:

“A single-point anchor with unloading grooves was adopted to reduce external stress coupled to the ZnO nano-resonator. (Detailed information was provided in supplementary materials Note S4)”

- ✓ Another detailed explanation was added in this revision on page 12~13:

The Q factor of the accelerometer is limited by the structure of the devices, which affects the resolution of the resonant accelerometer owing to the noise of the resonator. The high sensitivity of the ZnO resonant nano-accelerometer, however, could compensate the drawback of the Q factor partly. We will devote our energy to accomplish Q factor at least several thousands, which have achieved in our previous research³⁷, and a compensation algorithm based on the simulation and experiment in future study to solve this deficiency effectively.

- ✓ Another detailed explanation was added in supplementary materials Note S4:

Note S4 Design of the unloading grooves anchor

The differential resonant accelerometer can suppress the majority of the resonant frequency shifts caused by temperature, pressure and other factors. The size and properties of the two ZnO nanowires, however, cannot guaranteed to be completely consistent. So the resonant frequency fluctuations caused by temperature, pressure and other factors cannot eliminate completely. In the configuration involving a silicon structure, a ZnO nanowire, and a metal electrode in direct contact. Consequently, owing to distinct thermal expansion coefficients among these three materials, internal stresses are introduced as a consequence of thermal expansion. Specifically, axial stress induced by thermal expansion directly impacts the resonant frequency of the ZnO nano-resonator. To effectively address this phenomenon, a differential design of device structures and parameter optimization could eliminate the majority of the frequency shifts caused by structural expansions and internal stresses. Furthermore, the design of an anchor structure that releases stress promptly can also alleviate the impact of thermal stress on the ZnO nano-resonator. Figure S5 illustrates schematic diagrams depicting rectangular anchor points and unloading anchor points, respectively.

To investigate the influence of thermal stress on the three anchor points, the resonant frequency shifts of the ZnO resonant nano-accelerometer were analyzed using Finite Element Analysis. The operational temperature ranged from 25°C to 50°C. The Young's modulus values for ZnO and Si were assumed to be 150 GPa and 120 GPa, respectively. The coefficients of thermal expansion for ZnO and Si were obtained from sources such as D. Taylor *et al. Transactions and Journal of the British Ceramic Society*, v83, No. 1, p1, R.R. Rebber *et al. Journal of Applied Physics*, v41, No. 13, and C.A. Swenson *et. al, Journal of Physical and Chemical Reference Data*, v12, No. 2, within the specified temperature range. As illustrated in Figure S5, the differentially designed architecture successfully mitigated the

majority of frequency shifts attributed to temperature variations. Furthermore, the structure with unloading grooves contributed to the stability improvement by facilitating the release of stress resulting from temperature changes.

Figure S5. The anchor structure of accelerometer. (a) The rectangle anchor structure. (b) The unloading anchor structure. The resonant frequency shifts of the rectangle anchor structure (c) and the unloading anchor structure (d).

Comment 4:

Sensitivity is one of performance metrics for evaluation of accelerometer. The primary metrics are dynamic range and bandwidth. None of these are discussed / no data is presented.

Answers:

- ✓ We appreciate the insightful and valuable comments from the reviewer. In this revised version, the analysis of the device's dynamic range, bandwidth and bias instability was added. The

sensitivity of the ZnO resonant accelerometer was measured to be 16.818 kHz/g owing to the nanoscale ZnO resonant beam. In order to evaluate the accelerometer performance comprehensively, the dynamic range and bandwidth of accelerometer was evaluated with the open loop measurement, and the stability of the accelerometer was evaluated with Allan deviation based on the feedback loop measurement setup.

- ✓ The detailed explanation was added in this revision on page 9~10:

The resonant behaviors of the ZnO nano-resonator were actuated and detected electrically due to its transistor properties of semiconducting. An AC voltage was applied on the source, and a combination of AC voltage and DC voltage was applied on the gate to actuate the ZnO nano-resonator into high frequency vibration motion. To evaluate the resonant frequency of ZnO nano-resonator, a mixing principle based on lock-in amplifier was employed to read the frequency response of the ZnO nano-resonator, as shown in Figure 3b. The ZnO nano-resonator was placed in a vacuum environment under pressure of about 10^{-2} Torr to obtain higher quality factor (Q). The current frequency curve of the ZnO nano-resonator with different DC voltage applied on the gate was displayed in Figure 3c. It was evident that the peak currents flowing through the ZnO nanobeam raised with an increase of the DC voltage applied on the gate, which was attributed to the larger stretched towards the gate resulting from the DC voltage²⁴⁻²⁶. Meanwhile, the dynamic range of the resonator was evaluated to be 19.65 dB. Additionally, the resonant frequency and the quality factor (Q-factor) of the ZnO nano-resonator were extracted from Figure 3c with Lorentz fitting, which was displayed in Figure 3d. The resonant frequency increased monotonically with increasing DC voltage applied to the gate, resulting from the spring constant hardening due to the stretching of the ZnO nanobeam. It's suggesting that the initial stress of the ZnO nano-resonator was small

enough to work in high sensitivity detection. More importantly, the resonant frequency tuning range of the ZnO nano-resonator was 0.65 MHz to 0.91 MHz, which may be adopted to compensate for the resonant frequency deviation caused by the fabrication error and residual stress of the accelerator. The Q-factor of ZnO nano-resonator decreased as the DC voltage applied on the gate increased, which suggested the accelerator to operate with small DC voltage to obtain high accuracy resonant frequency and acceleration. From the ZnO nano-resonator's open-loop measurement, the bandwidth of the ZnO nano-resonator was calculated as $\Delta f_r = \frac{f_{res}}{Q} \cong 4.78 \square 26.94$ kHz, which depends on the DC voltage applied on the gate.

- ✓ Another detailed explanation was added in revision on page 11:

Since the acceleration is determined from resonant frequency of ZnO nanowires, the frequency stability is one of the crucial aspects of a resonant accelerometer. Using the same setup shown in Figure 3(a), stability measurements of the output frequency signals was performed for 30 minutes at an integration time of 200 ms. The device was mounted at 0g position, and working temperature was about 25 °C. As displayed in Figure 4d, the Allan deviation was plotted for the device working at room temperature, showing a bias instability of 13.13 ug at 1.2 s integration time. (Detail information was provided in supplementary materials Note S5)

Comment 5:

ZNO is a moderate piezoelectric material. I appreciate the nanowires fabrication aspect of the presented work. However, AlScN and HZO films can also be deposited with few nanometers thickness and patterned by electron-beam lithography to the same size of nanowires, while offering

significantly higher piezoelectric coefficient. Please justify the advantage of ZNO nanowire over alternative piezoelectric films with larger coupling.

Answers:

- ✓ We appreciate the insightful and valuable comments from the reviewer. In comparison to ZnO nanowires, both AlScN and HZO exhibit superior piezoelectric coefficients; however, the fabrication processes of AlScN and HZO thin films, as well as their device applications, differ from those involving ZnO nanowires. Those devices were prepared with multiple lithography and structure transfer, with the achievable resolution contingent upon lithography precision. The ZnO resonant nano-accelerometer was crafted using microscale “Top-down” method and “Bottom-up” synthesized ZnO nanowires, making its feature size dependent on the synthesized ZnO nanowires. Furthermore, comparing the three materials, the ZnO nanowires process the simplest stoichiometric, facilitating its mass production. Simultaneously, owing to its straightforward composition, controllable material purity, and simplified fabrication processes, it was convenient to prepare high purity and performance ZnO nanowires with simple methods, mild condition and low cost.

The wurtzite structure of ZnO nanowires is fully oxidized, imparting heightened resistance to oxidation. This property makes them particularly suitable for prolonged usage in accelerometers when combined with vacuum packaging. Furthermore, such endeavors could pave the way for the future integration of "Bottom-up" synthesized functional nanomaterials, including but not limited to carbon nanotubes, graphene, and metallic oxide nanowires, as key components in nanoelectronics. Once again, we express our gratitude for the reviewer's insightful comments, providing a novel perspective on resonant nano-accelerometers for us

and encouraging us to explore the AlScN and HZO films based force gauges to evaluate the acceleration in future research.

- ✓ The detailed explanation was added in this revision on page 13:

Meanwhile, the ZnO nanowire was totally oxidized owing to its wurtzite structure, which makes the ZnO resonant nano-accelerometer suitable for long term application with vacuum packaging, such as inertial navigation for robotics, unmanned aerial vehicles and wearable devices. What's more, such effort could encourage the combination of "top-down" methods with developed synthesized functional nanomaterials (i.e. carbon nanotubes, graphene, metal oxide nanostructures, *et. al.*), which would be an ideal substitute choice for silicon devices with performances could exceed from those.

Comments from Reviewer #3:

General comments

Comment 1:

Grammar, use of verb tenses, emphasis of sentences, all need a proof revision by a native or fluent English speaker.

Answers:

- ✓ We appreciate the insightful and valuable comments from the reviewer. In this revised version, we have conducted a thorough language review and modification of the manuscript to ensure the language is smooth and accurate. To further enhance the linguistic quality of the article, we invited Dr. Tiesheng Wang to edit and polish the paper, facilitating readers' comprehension of the content.

Comment 2:

At the basis of all authors claims is the use of ZnO for NEMS resonators as transduction mean, and the claim of the highest sensitivity in the literature. This is not true. Other works already used NEMS resonators, and reached larger sensitivity than the authors. E.g. Miani in proc. IEEE MEMS 2021 and later in proc. IEEE Inertial 2022 reached scale factors as large as 1.4 MHz/g which is two orders of magnitude larger than the present work. Authors need for sure to mention those works. □

Answers:

- ✓ We appreciate the insightful and valuable comments from the reviewer. We have read the relevant articles and learned more novel methods to fabricate NEMS resonant accelerometers.

Discussions about these devices were added in the introduction section. The relevant articles were also referred in this version, which are listed in the References.

- ✓ The detailed explanation was added in this revision on page 9~10:

Benefitting from nanomanufacturing technologies, the application of nano-resonators has promoted the resolution of the accelerometer to overcome the limitation of resolution-bandwidth-footprint trade-off in traditional technologies. Such accelerometer has been developed with a scale factor up to 1.4 MHz/g and a broad bandwidth of 1.5 kHz by Miani *et al.*^{25,26}.

- ✓ The added references are listed as follows:

25. T. Miani, *et al.*, Resonant accelerometers based on nanomechanical piezoresistive transduction, 34th IEEE MEMS, Electr Network, 192-195, (2021)

26 T. Miani, *et al.*, Nanoresonator-based accelerometer with large bandwidth and improved bias stability, 9th IEEE INERTIAL, Avignon, France, 1-4, (2022)

Comment 3:

Additionally, those results were obtained with monolithic Silicon structures (also NEMS gauges were made of Si, with 250 nm width), with no need of extra steps and extra materials. Authors should mention, indeed, the drawbacks of mixing different materials with different CTE and TCE in terms of overall system stability.

Answers:

- ✓ We appreciate the insightful and valuable comments from the reviewer. In this revised version, the resonant frequency fluctuation caused by temperature change was analyzed. Due to the change of the working temperature in a long period of time, the resonant frequency of the

nano-resonators would shift, which limits the accelerometer measurement accuracy and repeatability. The different coefficient of thermal expansion (CTE) of the mixing different materials makes the temperature frequency coefficient (TCF) changed with the temperature. To overcome the drawbacks resulted from the mixing different materials, differential configuration and unloading grooves anchor structure were adopted, which facilitate the elimination of stress induced by temperature, pressure and other external factors.

- ✓ The detailed explanation was added in this revision on page 3:

A single-point anchor with unloading grooves was adopted to reduce external stress coupled to the ZnO nano-resonator. (Detailed information was provided in supplementary materials Note S4)

- ✓ Another detailed explanation was added in supplementary materials Note S4:

Note S4 Design of the unloading grooves anchor

The differential resonant accelerometer can suppress the majority of the resonant frequency shifts caused by temperature, pressure and other factors. The size and properties of the two ZnO nanowires, however, cannot guaranteed to be completely consistent. So the resonant frequency fluctuations caused by temperature, pressure and other factors cannot eliminate completely. In the configuration involving a silicon structure, a ZnO nanowire, and a metal electrode in direct contact. Consequently, owing to distinct thermal expansion coefficients among these three materials, internal stresses are introduced as a consequence of thermal expansion. Specifically, axial stress induced by thermal expansion directly impacts the resonant frequency of the ZnO nano-resonator. To effectively address this phenomenon, a differential design of device structures and parameter optimization could eliminate the majority of the frequency shifts caused by structural expansions and internal stresses.

Furthermore, the design of an anchor structure that releases stress promptly can also alleviate the impact of thermal stress on the ZnO nano-resonator. Figure S5 illustrates schematic diagrams depicting rectangular anchor points and unloading anchor points, respectively.

Figure S5. The anchor structure of accelerometer. (a) The rectangle anchor structure. (b) The unloading anchor structure. The resonant frequency shifts of the rectangle anchor structure (c) and the unloading anchor structure (d).

To investigate the influence of thermal stress on the three anchor points, the resonant frequency shifts of the ZnO resonant nano-accelerometer were analyzed using Finite Element Analysis. The operational temperature ranged from 25°C to 50°C. The Young's modulus values for ZnO and Si were assumed to be 150 GPa and 120 GPa, respectively. The coefficients of thermal expansion for ZnO and Si were obtained from sources such as D. Taylor *et al. Transactions and Journal of the British Ceramic Society*, v83, No. 1, p1, R.R. Rebber *et al. Journal of Applied Physics*, v41, No. 13, and C.A. Swenson *et. al, Journal of*

Physical and Chemical Reference Data, v12, No. 2, within the specified temperature range. As illustrated in Figure S5, the differentially designed architecture successfully mitigated the majority of frequency shifts attributed to temperature variations. Furthermore, the structure with unloading grooves contributed to the stability improvement by facilitating the release of stress resulting from temperature changes.

Comment 4:

Characterization is absolutely preliminar. Accelerometers have been on the market for 30 years, and just showing a rough measure of the scale factor with only a few points is definitely not enough. Any modern paper on accelerometers shall show (i) scale factor up to the full-scale range and corresponding nonlinearity, (ii) noise in terms of PSD or even better in terms of root Allan variance and possibly also (iii) temperature and vibration stability. It is only the combination of these parameters that tells you whether an accelerometer is suited or not for specific applications. Just boosting the scale factor, e.g., might simply lower the full-scale range and make the device inadequate for several applications.

Answers:

- ✓ We appreciate the insightful and valuable comments from the reviewer. In this revised version, the characterization of the ZnO resonant nano-accelerometer was improved, where the bandwidth, scale factor and Allan variance were provided. With the open loop measurement of the ZnO nano-resonators, the bandwidth was calculated to be from 4.78 to 26.94 kHz, which depends on the DC voltage applied on the gate. With the feedback loop measurement, the resonant frequency of ZnO nano-resonators was traced when the device was working at room temperature. The Allan variance curve was prepared with the resonant frequency differential trace as function of time, which yields a bias instability of 13.13 μg at 1.2 s integration time.

- ✓ The detailed explanation was added in this revision on page 10:

From the ZnO nano-resonator's open-loop measurement, the bandwidth of the ZnO nano-resonator was calculated as $\Delta f_r = \frac{f_{res}}{Q} \cong 4.78 \square 26.94$ kHz, which depends on the DC voltage applied on the gate.

- ✓ Another detailed explanation was added in this revision on page 11:

Since the acceleration is determined from resonant frequency of ZnO nanowires, the frequency stability is one of the crucial aspects of a resonant accelerometer. Using the same setup shown in Figure 3(a), stability measurements of the output frequency signals was performed for 30 minutes at an integration time of 200 ms. The device was mounted at 0g position, and working temperature was about 25 °C. As displayed in Figure 4d, the Allan deviation was plotted for the device working at room temperature, showing a bias instability of 13.13 μ g at 1.2 s integration time. (Detail information was provided in supplementary materials Note S5)

Minor comments

Comment 5:

Please complete the literature review with the works mentioned above and other recent works showing performance much better than those presented here.

Answers:

- ✓ We appreciate the insightful and valuable comments from the reviewer. We have read the relevant articles and learned more novel methods to fabricate NEMS resonant accelerometers.

Discussions about these devices were added in the introduction section. The relevant articles were also referred in this version, which are listed in the References.

- ✓ The detailed explanation was added in this revision on page 2:

In recent years, various materials and structures have been employed to enhance the sensitivity and stability of M/NEMS resonant accelerometers. A concept of micromechanical silicon oscillating accelerometer (SOA) was first proposed and an in-plane vibration electrostatically driven resonant accelerometer was successfully developed with a resonant frequency of 20 kHz and a scale factor of 100 Hz/g using silicon-on-glass (SOG) processes by Gubbon *et al.*^{9,19}. Comi *et al* have developed a two-axis single-stage lever resonant accelerometer with a resonant frequency of 15 kHz and a sensitivity of 250 Hz/g^{20,21}. To obtain higher stability and sensitivity, Pandit *et al.* have designed and demonstrated a single-stage amplified differential resonant accelerometer with a resonant frequency exceeding 100 kHz and a differential acceleration scale factor of 5.61 kHz/g for high precision acceleration²²⁻²⁴. Benefitting from nanomanufacturing technologies, the application of nano-resonators has promoted the resolution of the accelerometer to overcome the limitation of resolution-bandwidth-footprint trade-off in traditional technologies. Such accelerometer has been developed with a scale factor up to 1.4 MHz/g and a broad bandwidth of 1.5 kHz by Miani *et al.*^{25,26}. The doped Si nano-resonator is, however, vulnerable to oxidation which will be inevitably detrimental for long-term application even with vacuum packaging. Villanueva *et al.* have raised a resonant NEMS accelerometer based on graphene nano-resonator which happens to resolve this issue with a sensitivity of approximately 4.41 kHz/g and a resonant frequency of 40 kHz²⁷. However, the resonant frequency measurements of the graphene

devices were performed by using a laser doppler vibrometer (LDV) points at the device and excited by a piezo-shaker, which were not compatible for integration with electronic circuits.

✓ The added references are listed as follows:

9 Hopkins, R., Miola, J. & Setterlund, R. The silicon oscillating accelerometer: A high-performance MEMS accelerometer for precision navigation and strategic guidance applications. *proceedings of annual meeting of the institute of navigation*, (2006).

20 Comi, C. *et al.* A Resonant Microaccelerometer With High Sensitivity Operating in an Oscillating Circuit. *Journal of Microelectromechanical Systems* **19**, 1140-1152, (2010).

21 Caspani, A., Comi, C., Corigliano, A., Langfelder, G. & Tocchio, A. Compact biaxial micromachined resonant accelerometer. *Journal of Micromechanics and Microengineering* **23**, 1-11, (2013).

22 Xudong, Z., Thiruvengatanathan, P. & Seshia, A. A. A Seismic-Grade Resonant MEMS Accelerometer. *Journal of Microelectromechanical Systems* **23**, 768-770, (2014).

23 Zou, X. & Seshia, A. A. Non-Linear Frequency Noise Modulation in a Resonant MEMS Accelerometer. *IEEE Sensors Journal* **17**, 4122-4127, (2017).

24 Pandit, M., Zhao, C., Sobreviela, G., Zou, X. & Seshia, A. A High Resolution Differential Mode-Localized MEMS Accelerometer. *Journal of Microelectromechanical Systems* **28**, 782-789, (2019).

25. T. Miani, *et al.*; Resonant accelerometers based on nanomechanical piezoresistive transduction, 34th IEEE MEMS, Electr Network, 192-195, (2021)

26 T. Miani, *et al.* Nanoresonator-based accelerometer with large bandwidth and improved bias stability, IEEE INERTIAL, Avignon, France, 1-4, (2022)

Comment 6:

No mention is given to the packaging, which instead has a major role in state-of-the-art performance of accelerometers, especially in terms of stability vs temperature, bending ecc...

Answers:

- ✓ We appreciate the insightful and valuable comments from the reviewer. Packaging plays a critical role in ensuring the optimal performance of ZnO resonant accelerometers, aiming to establish an ideal working environment and mitigate the impact of external factors such as temperature and pressure. Simultaneously, the vacuum environment plays a crucial role in enhancing the Q factor of ZnO resonant accelerometers, a key parameter for achieving high-precision acceleration measurements.

In this study, the accelerometer was securely housed in a ceramic package and connected through wire bonding to the electrodes on the printed circuit board (PCB). The PCB was installed on a rotary table, allowing for the generation of different acceleration levels by adjusting the angle between the plate and the horizontal line. To actuate and detect the ZnO nano-resonator, characterized by a high Q factor, a vacuum probe station was employed to create an environment with a pressure of approximately 10^{-2} Torr. In order to improve the performance of the device, our future efforts will be dedicated to focusing on the vacuum packaging process and developing a stability compensation algorithm in the future.

Comment 7:

The claim that "NEMS accelerometers are extensively used" is untrue. Most of the published accelerometers are MEMS based, while NEMS based are only a few.

Answers:

- ✓ We appreciate the insightful and valuable comments from the reviewer. In this revised version, the expression about the application of the accelerometer was modified to improve the manuscript's rigorosity and accuracy. Recently, MEMS accelerometers were wide applied in consumer electronics, industrial equipment and scientific exploration due to due to their small size, low power consumption, and high reliability. NEMS accelerometers were expected to be used in high precision measurement owing to the more miniaturized beam resonators acting as ultrasensitive force sensors. We would like to express our gratitude again for the reviewer's insightful comments, improving the accuracy of expression and avoiding misunderstanding to readers.

- ✓ The detailed explanation was added in this revision on page 2:

Microelectromechanical systems (MEMS) accelerometers have been extensively used in the geophysics¹⁻³, consumer electronics⁴⁻⁷, and navigation systems^{8,9} due to their small size, low power consumption, and high reliability. It is, however, still challenging to build high performance accelerometer to achieve high sensitivity and stability^{10,11}. Both noise reduction and sensitivity enhancement to external acceleration are required for such high performance device. Nanoelectromechanical systems (NEMS) resonant accelerometer converts the external acceleration to a modulated frequency output by changing the stiffness of a resonator which is compatible to integrated circuits and high resistant to external electronic noises¹²⁻¹⁶. NEMS resonant accelerometer is expected to have a strong competence in high precision acceleration measurement owing to its ultrasmall mass, high resonant frequency and ultralow power consumption^{17,18}.

Comment 8:

Once again, the increase in sensitivity is not the fundamental parameter for high precision inertial sensors, as long as it is not accompanied by other improvements (noise, wide FSR).

Answers:

✓ We appreciate the insightful and valuable comments from the reviewer. In this revised version, the discussion about the ZnO resonant nano-accelerometer's bandwidth and bias instability were added to demonstrate the device's improvement. The sensitivity of ZnO resonant nano-accelerometer was demonstrated to be 16.81 kHz/g, benefitting from the ZnO nano-resonator. The bandwidth of the accelerometer was evaluated to be from 4.78 to 26.94 kHz, depending on the voltage applied on the gate. The resonant frequency fluctuation of the accelerometer was traced using a feedback loop based on a lock-in amplifier. The Allan deviation of the resonant frequency fluctuation yields a bias instability of 13.13 μg at 1.2 s integration time when the device was working at room temperature.

✓ The detailed explanation was added in this revision on page 10:

From the ZnO nano-resonator's open-loop measurement, the bandwidth of the ZnO nano-resonator was calculated as $\Delta f_r = \frac{f_{res}}{Q} \cong 4.78 \sim 26.94 \text{ kHz}$, which depends on the DC voltage applied on the gate.

✓ Another detailed explanation was added in this revision on page 11:

Since the acceleration is determined from resonant frequency of ZnO nanowires, the frequency stability is one of the crucial aspects of a resonant accelerometer. Using the same setup shown in Figure 3(a), stability measurements of the output frequency signals was performed for 30 minutes at an integration time of 200 ms. The device was mounted at 0g

position, and working temperature was about 25 °C. As displayed in Figure 4d, the Allan deviation was plotted for the device working at room temperature, showing a bias instability of 13.13 ug at 1.2 s integration time. (Detailed information was provided in supplementary materials Note S5)

Comment 9:

The claim that resonant accelerometers are not vulnerable to electronic noise is wrong. Phase noise arises from the electronic circuit and affects the performance. Unfortunately, authors were not able to predict the noise limits and no noise measurements were shown in the manuscript.

Answers:

- ✓ We appreciate the insightful and valuable comments from the reviewer. In this revised version, the expression about the differential feature was changed to improve its rigorosity and accuracy. The differential resonant accelerometer has some advantages in static noise, and can suppress the resonant frequency shifts caused by temperature, pressure and other factors. The size and properties of the two ZnO resonators, however, cannot be guaranteed to be completely consistent. In order to minimize the internal stress of the ZnO nano-resonators, an unloading grooves anchor was adopted to facilitate releasing the stress caused by temperature, pressure and other factors. Additionally, a feedback loop was created to trace the accelerometer's resonant frequency fluctuation at about 25 °C, when the device was mounted at 0g position. The Allan deviation was employed to evaluate the device's bias instability.
- ✓ Another detailed explanation was added in this revision on page 3:

As a foundation for its high performance, a pair of ZnO nanowires were fixed on the proof mass which swung around a notched flexure when subjected to external acceleration. This

leads to a change in axial stress in the symmetrically distributed ZnO nanowires resulting in a shift in the resonant frequency. Compared to the single resonator design, the prototyped device evaluated the acceleration with a differential architecture, which could eliminate the majority of the resonant frequency shifts caused by temperature changes intrinsically.

- ✓ Another detailed explanation was added in this revision on page 3:

A single-point anchor with unloading grooves was adopted to reduce external stress coupled to the ZnO nano-resonator. (Detailed information was provided in supplementary materials Note S4)

- ✓ Another detailed explanation was added in this revision on page 11:

Since the acceleration is determined from resonant frequency of ZnO nanowires, the frequency stability is one of the crucial aspects of a resonant accelerometer. Using the same setup shown in Figure 3(a), stability measurements of the output frequency signals was performed for 30 minutes at an integration time of 200 ms. The device was mounted at 0g position, and working temperature was about 25 °C. As displayed in Figure 4d, the Allan deviation was plotted for the device working at room temperature, showing a bias instability of 13.13 ug at 1.2 s integration time. (Detailed information was provided in supplementary materials Note S5)

- ✓ Another detailed explanation was added in supplementary materials Note S4:

Note S4 Design of the unloading grooves anchor

The differential resonant accelerometer can suppress the majority of the resonant frequency shifts caused by temperature, pressure and other factors. The size and properties of the two ZnO nanowires, however, cannot guaranteed to be completely consistent. So the resonant

frequency fluctuations caused by temperature, pressure and other factors cannot eliminate completely. In the configuration involving a silicon structure, a ZnO nanowire, and a metal electrode in direct contact. Consequently, owing to distinct thermal expansion coefficients among these three materials, internal stresses are introduced as a consequence of thermal expansion. Specifically, axial stress induced by thermal expansion directly impacts the resonant frequency of the ZnO nano-resonator. To effectively address this phenomenon, a differential design of device structures and parameter optimization could eliminate the majority of the frequency shifts caused by structural expansions and internal stresses. Furthermore, the design of an anchor structure that releases stress promptly can also alleviate the impact of thermal stress on the ZnO nano-resonator. Figure S5 illustrates schematic diagrams depicting rectangular anchor points and unloading anchor points, respectively.

Figure S5. The anchor structure of accelerometer. (a) The rectangle anchor structure. (b) The unloading anchor structure. The resonant frequency shifts of the rectangle anchor structure (c) and the unloading anchor structure (d).

To investigate the influence of thermal stress on the three anchor points, the resonant frequency shifts of the ZnO resonant nano-accelerometer were analyzed using Finite Element Analysis. The operational temperature ranged from 25°C to 50°C. The Young's modulus values for ZnO and Si were assumed to be 150 GPa and 120 GPa, respectively. The coefficients of thermal expansion for ZnO and Si were obtained from sources such as D. Taylor *et al. Transactions and Journal of the British Ceramic Society*, v83, No. 1, p1, R.R. Rebber *et al. Journal of Applied Physics*, v41, No. 13, and C.A. Swenson *et. al, Journal of Physical and Chemical Reference Data*, v12, No. 2, within the specified temperature range. As illustrated in Figure S5, the differentially designed architecture successfully mitigated the majority of frequency shifts attributed to temperature variations. Furthermore, the structure with unloading grooves contributed to the stability improvement by facilitating the release of stress resulting from temperature changes.

Comment 10:

Which is the target application for your device? Authors mention in a generic way next generation devices, but it is impossible to conceive a device for all applications! Device design shall be indeed application driven, so authors shall specify the target application and related aforementioned parameters (FSR, noise, linearity, bandwidth...).

Answers:

- ✓ We appreciate the insightful and valuable comments from the reviewer. In this revised version, the potential application of the ZnO resonant nano-accelerometer was discussed. The sensitivity, bandwidth and bias instability of the device was demonstrated to be 16.818 kHz/g, from 4.78 to 26.94 kHz and 13.13 μg at 1.2 s integration time, respectively. Combined with the high resistant oxidation of ZnO nanowires, the ZnO resonant nano-accelerometer was

expected to be suitable for long term application for inertial navigation for robotics, unmanned aerial vehicles and wearable devices.

- ✓ The detailed explanation was added in this revision on page 13:

Meanwhile, the ZnO nanowire was totally oxidized owing to its wurtzite structure, which makes the ZnO resonant nano-accelerometer suitable for long term application with vacuum packaging, such as inertial navigation for robotics, unmanned aerial vehicles and wearable devices. What's more, such effort could encourage the combination of "top-down" methods with developed synthesized functional nanomaterials (*i.e.* carbon nanotubes, graphene, metal oxide nanostructures, *et. al*), which would be an ideal substitute choice for silicon devices with performances could exceed from those.

Comment 11:

Why are sensitivity predictions made only via FEA? There is a rather simple mechanical calculation to predict the scale factor...

Answers:

- ✓ We appreciate the insightful and valuable comments from the reviewer. In this revised version, the calculation formula of the accelerometer sensitivity was added, and the relationship between the sensitivity of the ZnO resonant accelerometer and the structure of the ZnO nano-resonator was analyzed. The accelerometer sensitivity with or without leverage, the influence of electrostatic excitation on the accelerometer sensitivity and the cross sensitivity of the ZnO nano-accelerometer in Y-direction and Z-direction were estimated by finite element analysis.
- ✓ The detailed explanation was added in this revision on page 7:

The resonant frequency of the ZnO nano-resonator under different acceleration, f_{res} , was described by Equation 1, where f_0 was the resonant frequency, m was the proof mass, a was the acceleration, I was the moment of inertia, d was the diameter, L was the length, ρ was the density and E was Young modulus of the nanowire, respectively. The resonant frequency differential of the two ZnO nano-resonators to the acceleration can be expanded as Equation 2, which indicates the resonant frequency shifts of the resonant accelerometer is proportional to the acceleration applied on the proof mass³⁰. Furthermore, the diameter and the length of the nanowire play great effect on the axial force sensitivity (Detailed information was provided in supplementary materials Note S1).

$$\left\{ \begin{array}{l} f_{res} = \frac{4.73^2}{2\pi} \sqrt{1 \pm \frac{0.749maL^2}{Ed^4}} \\ f_0 = \frac{4.73^2 d}{2\pi L} \sqrt{\frac{5E}{96\rho}} \end{array} \right. \quad (1)$$

$$\Delta f = 1.91 \frac{maL}{Ed^3} \sqrt{\frac{E}{\rho}} \quad (2)$$

- ✓ Another detailed explanation was added in supplementary materials Note S1:

Note S1. Principle of the resonant accelerometer

Resonant accelerometers use the characteristic of frequency drift of sensing elements under axial force to detect acceleration. The commonly used theoretical research methods for sensitive beams of resonant accelerometers are the Euler-Bernoulli beam theory and the Timoshenko beam theory. The Timoshenko beam theory considers the influence of beam cross-sectional rotation and shear deformation on the basis of the Euler-Bernoulli theory, and has significant improvement in the analysis results for non-slim beams and high-order modes. For slim beams with length-to-diameter ratio greater than 100 and ignoring the effects of

shear and rotation, the Euler-Bernoulli beam theory is more accurate in solving. Based on the vibration characteristics of Euler-Bernoulli beams, this section solves the resonant frequency of ZnO nanobeams and obtains the force sensitivity expression of the resonant accelerometer, and analyzes the nonlinear factors affecting the sensitivity of the resonant beam.

Resonant frequency of nanobeam

The ZnO nanobeam in the accelerometer with a large aspect ratio conforms to the typical Euler-Bernoulli beam. According to the classical bending vibration theory of Euler-Bernoulli beam, the free vibration equation of ZnO nanobeam shown in Figure S1 is:

Figure S1 Dynamic model of beam bending vibration

$$EI \frac{\partial^4 Y(z, t)}{\partial z^4} + \rho A \frac{\partial^2 Y(z, t)}{\partial t^2} = f(z, t) \quad (S1)$$

Where, I represents the moment of inertia of the resonant beam, E represents the Young's modulus of the ZnO resonant beam, ρ represents the density of the resonant beam, and A represents the cross-sectional area of the resonant beam.

According to the principle of superposition of vibration modes, the lateral displacement $Y(z, t)$ can be written as:

$$Y(z,t) = y(z)q(t) \quad (S2)$$

Where $y(z,t)$ is the lateral bending mode function, $q(t)$ is the propagation equation of the transverse wave, and $q(t)$ is harmonic. Substituting into equation S1 gives:

$$EI \frac{y''''(z)}{y(z)} - \frac{\rho A q''(t)}{q(t)} = \rho A \omega^2 \quad (S3)$$

Write the mode shape and time function into independent equations:

$$\begin{aligned} EI y''''(z) - \rho A \omega^2 y(z) &= 0 \\ q''(t) + \omega^2 q(t) &= 0 \end{aligned} \quad (S4)$$

The first equation in equation (S4) is a fourth-order differential equation, and its general solution is:

$$y(z) = C e^{sz} \quad (S5)$$

Substituting the mode shape function gives:

$$s^4 - \frac{\rho A}{EI} \omega^2 = 0 \quad (S6)$$

Let:

$$\beta^4 = \frac{\rho A \omega^2}{EI} \quad (S7)$$

The four solutions to the one-variable fourth-degree equation of equation (S7) are:

$$s_{1,2,3,4} = \pm \beta i, \pm \beta \quad (S8)$$

Therefore, the solution to the vibration mode function in equation (S4) is:

$$\begin{aligned} y(z) &= C_1 e^{-i\beta z} + C_2 e^{i\beta z} + C_3 e^{-\beta z} + C_4 e^{\beta z} \\ &= A \sin \beta z + B \cos \beta z + C \sinh \beta z + D \cosh \beta z \end{aligned} \quad (S9)$$

Where A, B, C, D are unknowns, and due to the fixed support at both ends of the ZnO nanobeam resonator, according to the boundary conditions of the fixed beam, the following equations can be obtained about the unknowns A, B, C, and D:

$$y(0) = 0, y'(0) = 0, y(L) = 0, y'(L) = 0 \quad (\text{S10})$$

$$\begin{pmatrix} 0 & 1 & 0 & 1 \\ \beta & 0 & \beta & 0 \\ \sin(\beta L) & \cos(\beta L) & \sinh(\beta L) & \cosh(\beta L) \\ \beta \cos(\beta L) & -\beta \sin(\beta L) & \beta \cosh(\beta L) & \beta \sinh(\beta L) \end{pmatrix} \begin{pmatrix} A \\ B \\ C \\ D \end{pmatrix} = \begin{pmatrix} 0 \\ 0 \\ 0 \\ 0 \end{pmatrix} \quad (\text{S11})$$

Since the coefficient matrix of equation (S11) cannot be all 0, the determinant of the coefficient matrix is 0, which is expressed as:

$$\begin{vmatrix} 0 & 1 & 0 & 1 \\ \beta & 0 & \beta & 0 \\ \sin(\beta L) & \cos(\beta L) & \sinh(\beta L) & \cosh(\beta L) \\ \beta \cos(\beta L) & -\beta \sin(\beta L) & \beta \cosh(\beta L) & \beta \sinh(\beta L) \end{vmatrix} = 0 \quad (\text{S12})$$

From equation (S12), the transverse free vibration frequency equation of the ZnO nanobeam can be obtained as:

$$\cos(\beta L) \cosh(\beta L) = 1 \quad (\text{S13})$$

Equation (S13) is a transcendental equation. The numerical solution of $\beta_i L$ can be obtained by numerical calculation methods, which are $\beta_1 L = 4.730$, $\beta_2 L = 7.853$, $\beta_3 L = 10.996$. Combined with equation (S7), the resonant frequency of the ZnO nanobeam in the transverse direction can be obtained as follows:

$$f_0 = \frac{(\beta_i L)^2}{2\pi} \sqrt{\frac{EI}{\rho AL^2}} \quad (\text{S14})$$

The ZnO nanobeam used in the accelerometer structure has a hexagonal cross section, and its moment of inertia and cross-sectional area are:

$$I = \frac{5\sqrt{3}d^4}{256}, A = \frac{3\sqrt{3}d^2}{8} \quad (\text{S15})$$

Where d is twice the edge length of the beam section. Under the microscope, the cross-section of the ZnO nanobeam is nearly circular. For ease of measurement, the parameter d in the following text refers to the diameter of the nanobeam, and L is the length of the resonant beam. Combined with equations (S8), (S14), and (S15), the first-order bending fundamental frequency of the ZnO nanobeam in free vibration can be obtained as follows:

$$f_0 = \frac{(\beta_1 L)^2}{2\pi} \sqrt{\frac{EI}{\rho AL^2}} \quad (\text{S16})$$

Then, the vibration mode function can be written as:

$$y(z) = \cos \beta z - \cosh \beta z - \frac{\cos \beta L - \cosh \beta L}{\sin \beta L - \sinh \beta L} (\sin \beta z - \sinh \beta z) \quad (\text{S17})$$

Sensitivity of the resonant accelerometer

Figure S2 shows the dynamic model of a vibrating beam subjected to axial force. When a resonant beam is subjected to an axial force, the vibration equation is expressed as:

$$EI \frac{\partial^4 Y(z,t)}{\partial z^4} + N \frac{\partial^2 Y(z,t)}{\partial z^2} + \rho A \frac{\partial^2 Y(z,t)}{\partial t^2} = f(z,t) \quad (\text{S18})$$

Figure S2 Dynamic model of vibration beam subjected to axial force

where N is the equivalent axial force on the resonant beam in the accelerometer under acceleration load, combined with equation (S2), equation (S18) can be written as:

$$EI \frac{\partial^4 y}{\partial z^4} q + N \frac{\partial^2 y}{\partial z^2} q + \rho A y \frac{\partial^2 q}{\partial t^2} = f(z, t) \quad (\text{S19})$$

Integrating equation (S19) over the entire resonant beam, we obtain:

$$\left(\int_L^0 \rho A y^2 dz \right) \ddot{q} + \left(\int_L^0 EI \frac{\partial^4 y}{\partial z^4} y dz + \int_L^0 N \frac{\partial^2 y}{\partial z^2} y dz \right) q = 0 \quad (\text{S20})$$

From the above equation, the maximum elastic potential energy and maximum kinetic energy of the resonant beam are:

$$\begin{cases} U_{\max} = \frac{1}{2} \int_L^0 EI \frac{\partial^4 y}{\partial z^4} y dz + \frac{1}{2} \int_L^0 N \frac{\partial^2 y}{\partial z^2} y dz \\ T_{\max} = \frac{1}{2} \omega^2 \int_L^0 \rho A y^2 dz \end{cases} \quad (\text{S21})$$

According to the law of conservation of energy, the natural angular frequency of the resonant beam under axial force can be obtained as:

$$\omega = \sqrt{\frac{\int_L^0 EI \frac{\partial^4 y}{\partial z^4} y dz + \int_L^0 N \frac{\partial^2 y}{\partial z^2} y dz}{\int_L^0 \rho A y^2 dz}} \quad (\text{S22})$$

Substituting the vibration mode equation for free vibration, we obtain the frequency of the resonant beam as:

$$\left\{ \begin{aligned} f &= \frac{\omega}{2\pi} = f_0 \sqrt{1 + \frac{NL^2}{4\pi^2 EI}} = f_0 \sqrt{1 + \frac{0.749NL^2}{Ed^4}} \\ f_0 &= \frac{(4.73)^2 d}{2\pi L^2} \sqrt{\frac{5E}{96\rho}} \end{aligned} \right. \quad (\text{S23})$$

Expanding equation (S23) using Taylor series, we obtain:

$$f = f_0 \left(1 + \frac{1}{2} \left[\frac{0.749L^2}{Ed^4} N \right] - \frac{1}{8} \left[\frac{0.749L^2}{Ed^4} N \right]^2 + \frac{1}{16} \left[\frac{0.749L^2}{Ed^4} N \right]^3 - \frac{5}{128} \left[\frac{0.749L^2}{Ed^4} N \right]^4 + o(N^5) \right) \quad (\text{S24})$$

When the accelerometer is subjected to a 1g acceleration, the resonant beam subjected to axial tension has an increased stiffness and bending vibration frequency, and the resonant beam subjected to axial compression has a decreased stiffness and bending vibration frequency.

The frequency difference between the resonant beams on the left and right sides is:

$$\Delta f = f_1 - f_2 = f_0 \left(\frac{0.749L^2}{Ed^4} N + \frac{1}{8} \left[\frac{0.749L^2}{Ed^4} N \right]^3 + o(n^5) \right) \quad (\text{S25})$$

Ignoring the influence of higher-order small quantities on sensitivity, the sensitivity expression is therefore:

$$\Delta f = f_0 \frac{0.749L^2}{Ed^4} N + \frac{1}{8} f_0 \left[\frac{0.749L^2}{Ed^4} N \right]^3 \quad (\text{S26})$$

The first part of equation (S26) is the linear sensitivity of the accelerometer, and the latter part affects the nonlinearity of the accelerometer sensitivity. Based on the force-sensitive characteristic of the resonant accelerometer, the sensitivity is affected by the size of the axial force, and it increases with the increase of the axial force. Since the sensitivity expression has a third-order nonlinearity term, it can be ignored when the axial force is small. When the axial force is large, it seriously affects the linearity of the accelerometer sensitivity, limiting the range of the accelerometer. Therefore, a balance must be struck between sensitivity, range, and the mutually suppressive effect of frequency nonlinearity. When the accelerometer is operating, the acceleration load acts on the mass block, and the small stiffness of the flexible neck on the supporting beam causes an equivalent bending moment that causes the mass block to rotate around the flexible beam, and the rotation angle is related to the size of the flexible beam and the mass block. Therefore, the parameter N , which represents the axial force on the nanobeam, is related to the structure, size, mass block size, flexible neck structure, and distance between the two beams of the accelerometer, all of which have a significant impact on the sensitivity of the accelerometer.

Comment 12:

ZnO material parameters (CTE, TCE ecc...) shall be provided and their impact on a silicon-based structure with different parameters, when temperature changes, shall be given.

Answers:

- ✓ We appreciate the insightful and valuable comments from the reviewer. In this revised version, the resonant frequency shifts caused by temperature was analyzed using finite element analysis. The majority of the frequency was eliminated with the differential architecture of the accelerometer. As the structure of ZnO nano-resonators cannot be completely consistent, the

resonant frequency shifts of the symmetrically distributed ZnO nano-resonators cannot be eliminated totally, which lead to a different frequency temperature coefficient (TCF) of the two ZnO nano-resonators. In order to suppress the different TCF of the two ZnO nano-resonators, a single-point anchor with unloading grooves was adopted to release the thermal stress caused by temperature change. At the same time, according to the results of the simulation calculation, TCF is linear in the temperature range. Our subsequent work plan adopts experiments to verify the validity of the simulation calculation, and adopts the algorithm to compensate the resonant frequency offset caused by external factors such as temperature to eliminate the influence of temperature and other factors on the resonant frequency offset of the accelerometer.

- ✓ The detailed explanation was added in this revision on page 3:

As a foundation for its high performance, a pair of ZnO nanowires were fixed on the proof mass which swung around a notched flexure when subjected to external acceleration. This leads to a change in axial stress in the symmetrically distributed ZnO nanowires resulting in a shift in the resonant frequency. Compared to the single resonator design, the prototyped device evaluated the acceleration with a differential architecture, which could eliminate the majority of the resonant frequency shifts caused by temperature changes intrinsically.

- ✓ Another detailed explanation was added in this revision on page 3:

A single-point anchor with unloading grooves was adopted to reduce external stress coupled to the ZnO nano-resonator. (Detailed information was provided in supplementary materials Note S4)

- ✓ Another detailed explanation was added in supplementary materials Note S4:

Note S4 Design of the unloading grooves anchor

The differential resonant accelerometer can suppress the majority of the resonant frequency shifts caused by temperature, pressure and other factors. The size and properties of the two ZnO nanowires, however, cannot guaranteed to be completely consistent. So the resonant frequency fluctuations caused by temperature, pressure and other factors cannot eliminate completely. In the configuration involving a silicon structure, a ZnO nanowire, and a metal electrode in direct contact. Consequently, owing to distinct thermal expansion coefficients among these three materials, internal stresses are introduced as a consequence of thermal expansion. Specifically, axial stress induced by thermal expansion directly impacts the resonant frequency of the ZnO nano-resonator. To effectively address this phenomenon, a differential design of device structures and parameter optimization could eliminate the majority of the frequency shifts caused by structural expansions and internal stresses. Furthermore, the design of an anchor structure that releases stress promptly can also alleviate the impact of thermal stress on the ZnO nano-resonator. Figure S5 illustrates schematic diagrams depicting rectangular anchor points and unloading anchor points, respectively.

To investigate the influence of thermal stress on the three anchor points, the resonant frequency shifts of the ZnO resonant nano-accelerometer were analyzed using Finite Element Analysis. The operational temperature ranged from 25°C to 50°C. The Young's modulus values for ZnO and Si were assumed to be 150 GPa and 120 GPa, respectively. The coefficients of thermal expansion for ZnO and Si were obtained from sources such as D. Taylor *et al. Transactions and Journal of the British Ceramic Society*, v83, No. 1, p1, R.R. Rebber *et al. Journal of Applied Physics*, v41, No. 13, and C.A. Swenson *et. al, Journal of Physical and Chemical Reference Data*, v12, No. 2, within the specified temperature range.

As illustrated in Figure S5, the differentially designed architecture successfully mitigated the majority of frequency shifts attributed to temperature variations. Furthermore, the structure with unloading grooves contributed to the stability improvement by facilitating the release of stress resulting from temperature changes.

Figure S5. The anchor structure of accelerometer. (a) The rectangle anchor structure. (b) The unloading anchor structure. The resonant frequency shifts of the rectangle anchor structure (c) and the unloading anchor structure (d).

Comment 13:

The sole experimental measurements that are shown are rather poor: 5 points only to estimate the scalefactor, some of which are clearly out of the fitting line is not a good demonstration of a "high-precision" device. Instead, it seems very poor.

Answers:

- ✓ We appreciate the insightful and valuable comments from the reviewer. In this revised version, more measured resonant frequency of the ZnO resonant nano-accelerometer was added in evaluation of the device's sensitivity. Meanwhile, the reproductivity of ZnO resonant nano-accelerometers was demonstrated with check of the same device. The statistical diagram of devices' sensitivity prepared with same methods consolidates the advantaged of the ZnO nano-resonator.
- ✓ The detailed explanation was added in this revision on page 4:

Figure 4b displayed the typical frequency response of the ZnO nano-resonator under different acceleration, demonstrating a small variation in amplitude and signal-to-noise ratio. The resonant frequency shifts of the ZnO resonant nano-accelerometer under different accelerations were displayed in Figure 4c. The frequency shifts for the left and right ZnO nano-resonator were determined to be 8.416 kHz/g and 8.402 kHz/g, respectively. Since it is an accelerometer with a push-pull differential architecture (As displayed in Figure 1d), the differential frequency sensitivity was the sum of the resonant frequency shifts of the two ZnO nano-resonators (*i.e.* 16.818 kHz). As mentioned above, the resonant frequency of the ZnO nano-resonator was 0.715MHz, the relative frequency change of $\Delta f/f_n$ was 2.35×10^4 ppm. The non-linearity of each of ZnO nano-resonators and differential sensitivity were less than 1%. The sensitivity of the accelerometer was measured again after a month (16.298 kHz/g) demonstrating a good single device reproducibility. Additionally, over 20 accelerometers were replicated showing sensitivity over 10 kHz/g. The consistently high sensitivity consolidates the advantages of ZnO nano-resonators.

- ✓ Another detailed explanation was added in supplementary materials Note S6:

Note S6 Sensitivity reproducibility of ZnO resonant nano-accelerometers

The sensitivity of the accelerometer was checked many times, and the longest interval time was over a month. The sensitivity was measured to be 16.298 kHz/g, demonstrating a good single device reproducibility. What's more, over 20 ZnO resonant nano-accelerometers of were prepared by optical microscope manipulation and FIB technology. The sensitivity of the devices was measured as introduced in the manuscript, majority of them were up to 10 kHz/g, which benefited from the nanoscale ZnO resonators.

Figure S7 (a) The sensitivity of ZnO resonant nano-accelerometer after a month. (b) Sensitivity statistical diagram of ZnO resonant nano-accelerometers

REVIEWER COMMENTS

Reviewer #1 (Remarks to the Author):

The manuscript submitted by the authors contains sufficient information and fresh ideas. The outcomes were extensively discussed. I am satisfied with the author's response and this version can be accepted for publication.

Reviewer #2 (Remarks to the Author):

The authors have addressed all reviewer points in detail. The paper is now of much higher quality. The paper can be accepted in its current form.

Reviewer #4 (Remarks to the Author):

The presented manuscript is very interesting and the performances of the device are outstanding. The work is improved after the review process, but I think some points related to vibration characterization need to be further reviewed.

1) The authors make a lot of confusion about frequency shift and sensitivity. The first is measured in Hz and the second in Hz/g. These units have to be corrected over all the manuscript, figure and SI.

2) Regarding Allan variation measurement, the methodology is not correct. It has to be computed at the resonance frequency point and not on one side of the Lorentzian peak. Since using the peak it is not easy to understand if a change is related to increase or decrease of the frequency, normally phase signal is used instead of amplitude one. I suggest to recompute Allan deviation with open loop method using phase signal. Moreover with a 0.7 Mhz resonance frequency, sampling has to be much lower than 200 ms (at least below 1ms).

3) What is the meaning of the bandwidth reported? The reported data is the width of the Lorentzian peak that is not easy to compare with other device. Better to report Q factor.

4) Simulations have been performed up to 2g, while measurement up to 1 g. Why this discrepancy? It will be very useful, as asked by another reviewer to increase measurement range in order to understand which is the linear range of the resonators and when they deviate from linearity

5) In Fig.4c is complicated to understand which axis referrer to each measurement. I suggest also to superimposed to the points a linear connection to better read the graph

Comments from Reviewer #4:

Comment 1:

The authors make a lot of confusion about frequency shift and sensitivity. The first is measured in Hz and the second in Hz/g. These units have to be corrected over all the manuscript, figure and SI.

Answers:

- ✓ We appreciate the expert and valuable comments from the reviewer. In this revision, we have conducted a thorough review and modification of the manuscript, figure and SI to enhance the readability and quality of the manuscript. To avoid confusion about the frequency shift and sensitivity, the units of them was unified in Hz and Hz/g, respectively. The mismatched sentences were corrected and highlighted in the manuscript. We thank you again for your meticulous review and suggestions to improve the quality of the paper, and all units have been checked and corrected if needed.
- ✓ The detailed change was made in this revision on page 8, line 16~20:
The sensitivity of the ZnO resonant nano-accelerometers without microleverages was demonstrated to be 3.629 kHz/g, while the sensitivity of the ZnO resonant nano-accelerometer with microleverages was determined to be 11.687 kHz/g. The results indicated that the microleverage was helpful to improve the sensitivity of the resonant accelerometers.
- ✓ The detailed change was made in this revision on page 11, line 1~4:
The sensitivities of the left and right ZnO nano-resonator were determined to be 8.416 kHz/g and 8.402 kHz/g, respectively. Since it is an accelerometer with a push-pull differential architecture (As displayed in Figure 1d), the sensitivity of the device was the sum of the sensitivities of the two ZnO nano-resonators (*i.e.* 16.818 kHz/g).

- ✓ The detailed change was made in supplementary materials on page 12, line 1~7:

The accelerometer resistibility to temperature change was evaluated with the ratio of frequency shifts caused by temperature change to resonant frequency of accelerometers at 25 °C. As illustrated in Figure S5, the differentially designed architecture successfully mitigated the majority of frequency shifts attributed to temperature variations. Furthermore, the structure with unloading grooves contributed to the stability improvement by facilitating the release of stress resulting from temperature changes.

Comment 2:

Regarding Allan variation measurement, the methodology is not correct. It has to be computed at the resonance frequency point and not on one side of the Lorentzian peak. Since using the peak it is not easy to understand if a change is related to increase or decrease of the frequency, normally phase signal is used instead of amplitude one. I suggest to recompute Allan deviation with open loop method using phase signal. Moreover, with a 0.7 MHz resonance frequency, sampling has to be much lower than 200 ms (at least below 1ms).

Answers:

- ✓ We appreciate the expert and valuable comments from the reviewer. In this revision, Allan deviation obtained with open loop measurement was added to evaluate the vibration characterization of the ZnO nano-resonator. The device was actuated at the resonant frequency, and the signal output was recorded with a time constant of 1ms for 1800 s. The bias stability of nano-resonators was analysed with Allan variation.

The bias instability of resonant accelerometer, however, was usually evaluated with closed loop measurement directly^{1,2}. As the reviewer's suggestion, it was more accurate and scientific to trace the resonant frequency of the accelerometer at the point of the resonant frequency,

other than the side of the Lorentz peak. For resonators based on ZnO nanowires, however, the phase of the current flowing the resonator was also Lorentzian shape (as displayed in Figure R1), which made it was hard to trace the change of resonant frequency at accurate resonant point in real time. To overcome the deficiency, the feedback measurement strategy based on the value of current (or the value of current phase) was developed³. The bias stability of the accelerometer was evaluated directly with Allan deviation using the built up feedback loop measurement circuits.

Figure R1 The frequency curve of the ZnO nano-resonator (a) the current value, (b) the current phase.

- ✓ The detailed change was made in this revision on page 11, line 16~19:

As displayed in Figure 4d, the Allan deviation was plotted for the device working at room temperature, showing a bias instability of 13.13 μg at 1.2 s integration time, which accorded with the stability of the ZnO nano-resonator. (Detailed information was provided in supplementary materials Note S5)

- ✓ The detailed change was added in supplementary materials on page 12, line 8~18 and page 13, line 1~4:

The frequency stability of ZnO nano-resonator was one of the key metrics to reveal the noise level of the device, which determines the resolution of the ZnO nano-accelerometer. The

experiment Allan deviation of ZnO nano-resonators was measured in open loop measurement recording of the phase variation of electrical signal at resonant frequency. The ZnO nano-resonator was driven at its resonant frequency (716 kHz), and the signal output was recorded with a time constant of 1ms for 1800 s. Its stability was analysed with Allan variation as displayed in Figure S6. For acceleration measurement has to be focused on short integration time. Typically, we achieved an Allan deviation about of 10^{-6} for $\tau = 0.25$ s at room temperature.

Figure S6. Allan deviation of ZnO nano-resonator measured in open loop condition.

The open loop measurement method was able to calibrate the sensitivity of the ZnO nano-accelerometer, however, it was not suitable for the practical application. It was expected to build a closed loop to trace the resonant frequency in real time. So the feedback loop was prepared to trace the resonant frequency of the accelerometers with a lock-in amplifier. As displayed in Figure 3b, an AC voltage was applied on the source, and a combination of AC voltage and DC voltage was applied on the gate to actuate the ZnO nano-resonator into high frequency vibration motion. Meanwhile, a mixing principle based on lock-in amplifier was employed to read the frequency response of the ZnO nano-resonator. To trace the resonant

frequency, the current value was employed, keeping I around a reference value I_{ref} by varying f . As displayed in Figure S6, when I remains between I_{min} and I_{max} , the driving frequency was not changed and, accordingly, the shift of the resonant frequency was calculated from I and the slope of the frequency around I_{ref} . The feedback time can be made as low as 50 ms. The feedback is interrupted repeatedly (typically every 600 s) during ~10 s for a control of the lineshape of the current-frequency (I as a function of f): if the resonant lineshape significantly differs from that measured previous, the recorded data were discarded.

Comment 3:

What is the meaning of the bandwidth reported? The reported data is the width of the Lorentzian peak that is not easy to compare with other device. Better to report Q factor.

Answers:

- ✓ We appreciate the expert and valuable comments from the reviewer. In this paper, the Q factor of ZnO nano-accelerometer was measured and analysed, which was obtained with the quotient of resonant frequency and full width of half maximum (FWHM) of the current-frequency curve using Lorentz fitting. To improve the characterization of the ZnO resonant nano-accelerometer, the primary metrics (dynamic range and bandwidth) of accelerometers were also obtained with resonant frequency and Q factor of the device, which were acquired with the open loop measurement of the ZnO nano-accelerometer. Combined with the high resonant frequency of the ZnO nano-accelerometer, the resolution/bandwidth trade-off of traditional technology was overcome. Moreover, the Q factor limited by the ZnO nano-accelerometer structure was discussed in discussion part, and the future working schematic was arranged to improve the deficiency.

- ✓ The detailed explanation was in this revision on page 9, line 14~27:

Additionally, the resonant frequency and the Q factor of the ZnO nano-resonator were extracted from Figure 3c with Lorentz fittings, which were displayed in Figure 3d. The resonant frequency increased monotonically with increasing DC voltage applied to the gate, resulting from the spring constant hardening due to the stretching of the ZnO nano-resonator. It's suggesting that the initial stress of the ZnO nano-resonator was small enough to work in high sensitivity detection³⁴⁻³⁶. More importantly, the resonant frequency tuning range of the ZnO nano-resonator was 0.65 MHz to 0.91 MHz, which may be adopted to compensate for the resonant frequency deviation caused by the fabrication error and residual stress of the accelerometer. The Q factor of ZnO nano-resonator decreased as the DC voltage applied on the gate increased, which suggested the accelerometer to operate with small DC voltage to obtain high accuracy resonant frequency and acceleration. From the ZnO nano-resonator's open-loop measurement, the bandwidth of the ZnO nano-resonator was calculated as

$$\Delta f_r = \frac{f_{res}}{Q} \cong 4.78 \square 26.94 \text{ kHz, which depends on the DC voltage applied on the gate.}$$

- ✓ Another detailed explanation was in this revision on page 12, line 17~25:

The Allan deviation yields a bias instability of 13.13 μ g at 1.2 s integration time, while the Q factor of the accelerometer demonstrated to several hundred. The Q factor of the accelerometer is limited by the structure of the devices, which affects the resolution of the resonant accelerometer owing to the noise of the resonator. The high sensitivity of the ZnO resonant nano-accelerometer, however, could compensate the drawback of the Q factor partly. We will devote our energy to accomplish Q factor at least several thousands, which have achieved in our previous research³⁷, and a compensation algorithm based on the simulation and experiment in future study to solve this deficiency effectively.

Comment 4:

Simulations have been performed up to 2g, while measurement up to 1 g. Why this discrepancy? It will be very useful, as asked by another reviewer to increase measurement range in order to understand which is the linear range of the resonators and when they deviate from linearity

Answers:

- ✓ We appreciate the expert and valuable comments from the reviewer. It was helpful to understand the linear range of the accelerometer with increasing the measurement range. The sensitivity of high resolution accelerometer, however, was usually measured with tilt tests, making the measurement range of the accelerometer was limited to $\pm 1g^{4,5}$. The full-scale acceleration range and linearity errors of accelerometers were usually measured by exploiting the centripetal acceleration. Due to the limitations of experiment conditions, the centrifuge measurement of the ZnO resonant nano-accelerometer can't be carried out at present, and we are also trying to realize the necessary experimental conditions and equipment. The measurement results of the prototype ZnO nano-accelerometer, however, demonstrated the excellent performance of ZnO nano-accelerometers and the great potential of NEMS devices based on ZnO nanowires, carbon nanotubes et. al., which could be an ideal substitute choice for silicon devices with performances could exceed from those. We wish to express our gratitude again to reviewer for the insightful comment and suggestion, which help us to determine the future working schedule to improve the performance of the device.

Comment 5:

In Fig.4c is complicated to understand which axis referrer to each measurement. I suggest also to superimposed to the points a linear connection to better read the graph

Answers:

- ✓ We appreciate the expert and valuable comments from the reviewer. In this revision, the indicating arrow was added to reveal the relation between the measurement and axis, and the color of the figure was changed to enhance the readability of the figure. We thank you again for your meticulous review and suggestions, and we will ensure that all figures were clearly and easy understood.
- ✓ The detailed change was added in this revision on page 11:

Figure 4. Measurement setups and results of the ZnO resonant nano-accelerometer. (a) Schematic diagram of the accelerometer measurement. (b) The typical measurement results of resonant frequency of right beam of the differential resonant accelerometer under different external acceleration. (c) The resonant frequency and differential resonant frequency of the

ZnO resonant nano-accelerometer. (d) The Allan deviation of the ZnO resonant nano-accelerometer as the external acceleration was 0.

References

- [1] M. Pandit, A. Mustafazade, C. Zhao, G. Sobreviela, A. Seshia, An ultra-high resolution resonant MEMS accelerometer, 2019 IEEE 32nd International Conference on Micro Electro Mechanical Systems (MEMS), (2019).
- [2] H. Ding, C. Wu, J. Xie, A MEMS resonant accelerometer with high relative sensitivity based on sensing scheme of electrostatically induced stiffness perturbation. *Journal of Microelectromechanical Systems* 30, 32-41(2021)
- [3] J. Chaste, A. Eichler, J. Moser, G. Ceballos, R. Rurali, A. Bachtold, A nanomechanical mass sensor with yoctogram resolution. *Nature Nanotechnology* 7, 301-304 (2012)
- [4] C. Zhao, *et al*, A resonant MEMS accelerometer with 56ng bias stability and 98ng/Hz^{1/2} noise floor. *Journal of Microelectromechanical Systems* 28, 324-326 (2019)
- [5] T. Miani, *et al*, Resonant accelerometers based on nanomechanical piezoresistive transduction, 34th IEEE International Conference on Micro Electro Mechanical Systems (MEMS), 192-195 (2021)

REVIEWERS' COMMENTS

Reviewer #4 (Remarks to the Author):

Authors reply to my comments properly, therefore I suggest the publication. I would add a line to the point graph of Figure 4b for better clarity

Comments from Reviewer #4:

Comment 1:

Authors reply to my comments properly, therefore I suggest the publication. I would add a line to the point graph of Figure 4b for better clarity

Answers:

- ✓ We appreciate the expert and valuable comments from the reviewer. However, after careful consideration, we have decided not to incorporate the suggested addition of a line to the point graph of Figure 4b. We believe that the current presentation of Figure 4b, coupled with appropriate legends and annotations, effectively communicates the intended message to readers.